# Towards Reliable Offline Reinforcement Learning via Lyapunov Uncertainty Control

## Abstract

Learning trustworthy and reliable offline policies presents significant challenges due to the inherent uncertainty in pre-collected datasets. In this paper, we propose a novel offline reinforcement learning method to tackle this issue. Inspired by the concepts of Lyapunov stability and control-invariant sets from control theory, the central idea is to introduce a restricted state space for the agent to operate within. This approach allows the learned models to exhibit reduced Bellman uncertainty and make reliable decisions. To achieve this, we regulate the expected Bellman uncertainty associated with the new policy, ensuring that its growth trend in subsequent states remains within acceptable limits. The resulting method, termed Lyapunov Uncertainty Control (LUC), is shown to guarantee that the agent remains within a low-uncertainty state enclosure throughout its entire trajectory. Furthermore, we perform extensive theoretical and experimental analysis to showcase the effectiveness and feasibility of the proposed LUC.

## 1 Introduction

Offline reinforcement learning (RL) allows policy learning from historical data without real-world interaction. However, ensuring reliable sequential decision-making from offline data poses a significant challenge in practical applications. For example, in healthcare (Tang & Wiens, 2021), a reliable diagnostic agent requires avoiding unfamiliar approaches that may introduce errors in subsequent procedures. Similar requirements exist in fields such as autonomous driving (Kiran et al., 2022), robotics control (Lobbezoo et al., 2021), and others.

The reliability of offline RL is undermined by the risk of deviating from the scoped safe regions, i.e., stable safe control (Kang et al., 2022) from offline data. More precisely, our aim is to stop the offline-learned agent from entering areas that could cause severe consequences after deployment. Specifically, in practical applications, the safety requirements for decision making are extremely strict (Jiang et al., 2023), demanding that every decision made by the agent at each step be safe. Meanwhile, current methods like the pessimistic and DRRL methods fall short in handling this issue. For instance, Pessimistic methods such as MOPO (Yu et al., 2020), PBRL (Bai et al., 2022) and RORL (Yang et al., 2022) mainly concentrate on making the agent act in accordance with the demonstrations in the dataset. Although these methods are good at quantifying the OOD data, they may not entirely fulfill the previously mentioned safety requirements. On the other hand, Robust RL methods (Panaganti et al., 2022; Shi & Chi, 2024) aim to improve the agent's capacity to deal with distributional shift by establishing the uncertainty set of the transition function and optimizing the lower bound of the policy's long term returns under the worst case scenario within this set. Unfortunately, Robust RL also fails to take into account the safety issue described earlier. In other words, they are unable to prevent the learned agent from straying from the safe region.

Inspired by control-invariant sets in control theory (Kerrigan, 2000; Richter & Roy, 2017), where a closed region is delineated in the state space to offer a reliable working environment for the agent, a recent method named Lyapunov Density Model (Kang et al., 2022) defines a region based on data density distribution to ensure adequate data support for the agent during operation. However, setting a reliable region based on a common and pessimistic density criterion overlooks the issue of performance imbalances in complex environments, where data requirements for achieving a certain performance level may vary across regions. For example, in autonomous driving, data needs for learning on a smooth highway differ significantly from those on a rugged mountain road. Instead,

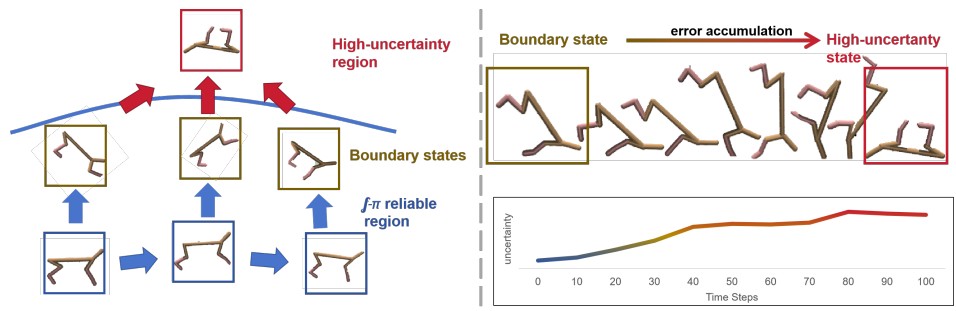

Figure 1: (left) Solely constraining current step's uncertainty is insufficient to identify those boundary states that pose a high risk of the agent deviating into high-uncertainty regions. (right) An illustration of the failure of traditional pessimism-based agents in accumulating errors by deviating from reliable regions, as demonstrated on a Halfcheetah robotic agent.

we advocate using a metric linked to model performance (e.g., value functions, policies) - Bellman uncertainty - for region definition to meet reliability standards in complex environments.

In particular, we aim to define a confined state space for the agent's operation, where the learned models demonstrate reduced Bellman uncertainty and reliable decision-making. To achieve this, we introduce a novel offline reinforcement learning method that regulates the expected Bellman uncertainty associated with the new policy. This regulation ensures that, at subsequent states, the growth trend of uncertainty remains within acceptable limits, allowing the agent to navigate low-uncertainty regions that serve as safe zones. Drawing inspiration from the control Lyapunov functions used in optimal control, we refer to our approach as Lyapunov Uncertainty Control (LUC). We implement our Lyapunov Uncertainty Control (LUC) method using a standard deviation-based uncertainty measure that relies on Q-ensembles, as described in Bai et al. (2022). Theoretically, we demonstrate that our approach can confine the learned agent to operate within a low-uncertainty state enclosure, resulting in secure and reliable trajectories. Furthermore, in certain scenarios, it can enhance the minimum performance bound of the new policy. Finally, we conduct extensive experiments to showcase the effectiveness and feasibility of LUC across various tailored benchmarks, including out-of-distribution (OOD) benchmarks and those with poor demonstrations.

The subsequent sections, after a brief review of related works, Section 3 present a concise overview of the fundamental concepts in offline RL. Section 4 elaborates on the LUC methodology, providing a detailed theoretical analysis of its effectiveness and implementation insights. Section 5 presents experimental results evaluating LUC's performance across various scenarios. To conclude, the paper summarizes the findings and contributions, along with a brief discussion on limitations.

## 2 RELATED WORKS

**Offline RL algorithms.**   Dealing with *distributional shift* poses a significant challenge for offline RL algorithms. Previous works, including CQL (Kumar et al., 2020), BEAR (Kumar et al., 2019), and BRAC (Wu et al., 2019), have aimed to tackle this issue by integrating conservative principles to prevent out-of-distribution (OOD) actions. However, these methods mainly concentrate on reducing the disparity between the new policy and the behavior policy that gathered the dataset. On the other hand, approaches like Implicit Q-Learning (IQL) (Kostrikov et al., 2022) entirely avoid OOD queries for actions during TD target estimation. Nonetheless, these methods heavily depend on the action distribution of the behavior policy, leading to a lack of generalization capability.

**Pessimistic offline RL.**   Pessimistic offline RL algorithms introduce Bellman uncertainty quantification (Jin et al., 2021; Xie et al., 2021) to determine reliable actions for generalizing to OOD data. This method has exhibited significant efficacy in model-based offline RL algorithms such as Model-based Offline Policy Optimization (Yu et al., 2020) and MOdel-Bellman Inconsistency penalized offLinE Policy Optimization (MOBILE)(Sun et al., 2023), as well as in model-free algorithms like Pessimistic Bootstrapping for offline RL (PBRL)Bai et al. (2022), Ensemble-Diversified Actor Critic (EDAC)An et al. (2021) and Robust Offline RL (RORL)(Yang et al., 2022). However, this

study reveals that solely managing uncertainty at the current step is insufficient to ensure reliability and safety, given the short-sightedness regarding the decision's potential outcomes.

**Consequence-driven offline RL.** Consequence-driven offline RL methods have been developed to address the *state distributional shift* issue utilizing the concept of *state recovery*(Zhang et al., 2022; Jiang et al., 2023). State Deviation Correction(Zhang et al., 2022) involves pre-training a forward dynamics model to facilitate the recovery process, whereas Out-of-sample State Recovery (OSR) (Jiang et al., 2023) employs an inverse dynamics model to implicitly execute the concept. While these techniques assist in rectifying the agent's behavior deviations from reliable regions, their unreliability in OOD states may hinder their success rates. In contrast, our LUC approach can be thought of as a more robust method which not only considers the reliability of the immediate consequence of executing a policy but that of its long term consequence.

**Robust Reinforcement Learning.** Robust RL's pessimism stems from penalizing with uncertainty in the outcome predictions of actions, to deal with the problem of distributional shift(Panaganti et al., 2022; Shi & Chi, 2024). However, if some behaviors that could lead to the deviation from the safe regions are supported well by dataset (as is in the case described in Figure 1), the penalty loses its effectiveness due to that the uncertainty in outcome predictions would be tiny, thereby exacerbating the risk of entering high-uncertainty regions. Therefore, we conclude that Robust RL are unable to prevent the learned agent from straying from the safe region, i.e., the safety requirements.

## 3 PRELIMINARIES

In the standard formulation of reinforcement learning, a Markov Decision Process (MDP) is used to model the problem. It is represented by a tuple $(S, A, P, R, \gamma, \rho_0)$, where $S$ denotes the state space, $A$ represents the action space, $M$ is the transition function (in a deterministic transitioned MDP, $M(s, a) = s'$, while in a stochastic transitioned MDP, $P(s'|s, a)$ is a distribution of states), $R$ is the reward function with upper bound $R_{max}$, $\gamma$ is the discount factor, and $\rho_0$ is the initial state distribution. A policy, denoted as $\pi : S \rightarrow A$, guides the decision-making process in interacting with the environment. To evaluate the expected cumulative rewards, a Q-value function $Q^\pi(s, a)$ is defined as $(1 - \gamma)\mathbb{E}[\sum_{t=0}^{\infty} \gamma^t R(s_t, \pi(a_t|s_t))|s, a]$. For convenience, the $\gamma$-discounted future state distribution (stationary state distribution) is defined as $d^\pi(s) = (1-\gamma)\sum t = 0^\infty \gamma^t Pr(s_t = s; \pi, \rho_0)$, with $\rho_0$ representing the initial state distribution and $(1 - \gamma)$ is the normalization factor.

In the offline setting, Q-Learning (Watkins & Dayan, 1992) is used to learn a Q-value function $\hat{Q}(s, a)$ and a policy $\pi$ from a dataset $\mathcal{D}$ collected by a behavior policy $\pi_\beta$. The dataset consists of quadruples $(s, a, r, s') \sim d^{\pi_\beta}(s)\pi_\beta(a|s)P(r|s, a)P(s'|s, a)$. The objective is to minimize the Bellman error over the offline dataset (Watkins & Dayan, 1992) and search for a good policy in the policy candidate set $\Pi \subset (S \rightarrow \triangle(A))$ under the supervision of a value-function class $\mathcal{F} \subset (S \times A \rightarrow [0, V_{max}])$ to model the Q-value function,

$$Q \longleftarrow \arg\min_Q \mathbb{E}_{(s,a,r,s')\sim\mathcal{D}}\big[r + \gamma[\max_{\pi\in\Pi}\mathbb{E}_{a'\sim\pi(\cdot|s')}Q(s', a')] - Q(s, a)\big]^2 \tag{1}$$

More specifically, in this paper, we denote the optimal Bellman operator over $\Pi$ as $\mathcal{T}^\Pi f(s, a) = r(s, a) + \gamma\mathbb{E}_{s'\sim P(\cdot|s,a)}[\max_{\pi\in\Pi}\mathbb{E}_{a'\sim\pi(\cdot|s')}Q(s', a')]$, and the empirical Bellman operator as $\hat{\mathcal{T}}^\Pi f(s, a) = r(s, a) + \gamma\mathbb{E}_{s'\sim\hat{P}(\cdot|s,a)}[\max_{\pi\in\Pi}\mathbb{E}_{a'\sim\pi(\cdot|s')}Q(s', a')]$, where $\hat{P}$ is the empirical dynamics model based on the dataset. It is worth noting that the TD target in Eq.(1) is estimated by the empirical Bellman operator.

## 4 METHOD

This section provides a detailed introduction to our work. In Sec.4.1, we formally define our objective mathematically as obtaining a policy that consistently operates within a reliable region. Subsequently, in Sec.4.2, we present a specific algorithm to accomplish this objective. Finally, in Sec.4.3, we analyze the theoretical properties of the algorithm, demonstrating its capability to improve the performance lower bound of the learned policy under specific scenarios.

### 4.1 OPERATING WITHIN RELIABLE REGIONS BY LYAPUNOV POLICY

In this section, we formally define our conceptual framework for Lyapunov Uncertainty Control. Specifically, analogous to control-invariant sets in control theory, we first define reliable regions in the state space where the policy $\pi$ can operate effectively. Previous methods defined these regions using density models (Kang et al., 2022); however, as mentioned earlier, a local region may not be reliable even with high density due to the complexity and nonlinearity of the underlying environment. Instead, we introduce a measurement based on epistemic uncertainty, denoted as $\zeta_f(s, a)$. The precise computational method for $\zeta_f(s, a)$ will be presented later; however, it is a positive scalar that is closely related to the agent's current knowledge, reflecting the generalization capability of the learned value function $f$ at the input data $(s, a)$. If $f$ generalizes well, $\zeta_f(s, a)$ will be small; conversely, if $f$ does not generalize well, $\zeta_f(s, a)$ will be large.

Using $\zeta_f(s, a)$, we can evaluate whether an induced policy $\pi$ can make reliable decisions at a given state $s$ by assessing the uncertainty of the learned value function $f$. Furthermore, we derive the following definition of the $f - \pi$ reliable region in Definition 1:

**Definition 1.** *($f - \pi$ reliable region). Given an arbitrary value function $f$, policy $\pi$ and a threshold $c$, we define the $f - \pi$ reliable region over the state space,*

$$\mathcal{G}_f(\pi) = \{s | \zeta_f(s, \pi) \le c\} \tag{2}$$

*where $\zeta_f(s, \pi) = \mathbb{E}_{a \sim \pi(\cdot | s)} \zeta_f(s, a)$.*

As shown in Figure 1, where 'low-uncertainty region' illustrates the agent's $f - \pi$ reliable regions. If the agent operates beyond its reliable regions, it would accumulate decision errors, finally failing the task. On the other hand, if an agent with policy $\pi$ always operate within its $f - \pi$ reliable regions, we call this policy a reliable policy, defined in Definition 2.

**Definition 2.** *(Reliable policy). A policy $\pi$ is reliable if it satisfies that $\forall s \in \mathcal{D} \cap \mathcal{G}_f(\pi)$, if $\forall t, s_t \in supp(P(s_t | s_0 = s, \pi))$, then $s_t \in \mathcal{G}_f(\pi)$.*

Furthermore, a Lyapunov policy, as defined in Definition 3, not only manages uncertainty at the current step but also addresses the tendencies of these uncertainty one step ahead. In other words, a Lyapunov policy is capable of controlling current step uncertainty to encompass reliable regions over the state space, while also restricting one-step forward uncertainty to ensure trajectory reliability.

**Definition 3.** *(Lyapunov policy). Given an arbitrary value function $f$ and offline dataset $\mathcal{D}$. A policy $\pi$ is a Lyapunov policy if it satisfies*

$$1. \forall s \in \mathcal{D}, \zeta_f(s, \pi) \le c; \qquad 2. \forall s \in \mathcal{D}, \max_{\hat{a} \in \pi} \zeta_f(M(s, \hat{a}), \pi) \le \zeta_f(s, \pi). \tag{3}$$

*where $M$ is the deterministic transition.*

Then we have the following results, shown in Theorem 1,

**Theorem 1.** *In a deterministic transitioned MDP, a Lyapunov policy $\pi$ is a reliable policy.*

*The proof of Theorem 1 is given in Appendix A.1.* Essentially, this theorem says that a Lyapunov policy is also a *Lyapunov reliable policy*. In other words, if a policy is a Lyapunov policy, then it will operate within its enclosed reliable region.

**Proposition 1.** *(Existence of reliable policy.) Suppose the dataset have a sufficient coverage over the optimal policy, i.e., $\sup_{s,a} \frac{\pi^*(a|s)}{\pi_\beta(a|s)} \le C^*$. Then there exists a reliable policy.*

*Proof of Proposition 1 could be seen in Appendix A.1.* Proposition 1 shows that there would always exist a reliable policy in the MDP system with sufficient data, which motivates us to learn such a policy for reliable control.

To summary, the main theoretical results in this section are used for problem formulation and functional requirements of the method - Definition 1 defines a safe region for the agent to operate, and Definition 2 defines reliable policy based on it, which is able to verify the safety requirements, i.e., operate within the region defined by Definition 1; Definition 3 and Theorem 1 indicate what kind of policies need to be learned to ensure that the agent can operate stably within the safe region without exceeding it. Proposition 1 demonstrates the existence of policies that meet these functional requirements, validating the feasibility of the method proposed in this paper.

## 4.2 IMPLEMENTING LYAPUNOV UNCERTAINTY CONTROL BY VALUE ESTIMATION

In this section, we use the Bellman uncertainty quantifier as in (Jin et al., 2021) to implement the value-epistemic uncertainty in Definition 1,

$$\zeta_f(s,a) = \|\mathcal{T}f - \hat{\mathcal{T}}f\|(s,a) \tag{4}$$

where $f$ is the learned value function. $\mathcal{T}$ is an arbitrary Bellman operator, while $\hat{\mathcal{T}}$ is its empirical version according to the dataset. Previous studies (An et al., 2021) suggest that Bellman uncertainty can rely on model predictions to evaluate state-action pairs. High variance in the model's prediction for a particular action implies inadequate data support, leading to low reliability. This property confirms that Bellman uncertainty aligns with the requirement in Definition 1.

Next, our objective is to acquire the Lyapunov reliable policy outlined in Definition 3 from the offline dataset using a model-free approach. Here, we present the Lyapunov value estimation, which straightforwardly penalizes not only the Q-value functions using the uncertainty quantifier from Eq.(4) at the current time step but also the increasing decision uncertainty tendency based on the next time step's situation, as,

$$\mathcal{L}_{LUC}(s,a,s',a',f) = \mathbb{E}_{a \sim \pi(\hat{a}|s)}\zeta_f(s,\hat{a}) + \beta \cdot (\zeta_f(s',a') - \zeta_f(s,a)) \tag{5}$$

Subsequently, we apply regularization using the offline dataset $\mathcal{D}$ and the learned models (Q function $f_k$ and new policy $\pi$), incorporating it as a penalty in the value estimation,

$$f_k(s,a) \leftarrow f_k(s,a) - \hat{\beta} \cdot \mathbb{E}_{a' \sim \pi(\cdot|s')}\mathcal{L}_{LUC}(s,a,s',a',f_k) \tag{6}$$

where $f_k$ is the Q function learned at the $k^{th}$ iteration. $\hat{\beta}$ is the balance coefficient. $(s,a,s')$ is the tuple sampled from the offline dataset $\mathcal{D}$, and the $\pi$ is the currently learned policy.

**Proposition 2.** *Suppose the action distribution of new policy $\pi(a|s)$ is positive correlated to the learnt Q function $f(s,a)$, i.e., $\pi(a|s) \propto f(s,a)$. Then the proposed Lyapunov value estimation in Eq.(5) induces a Lyapunov reliable policy as defined in Definition 3.*

The proof is available in Appendix A.1. Proposition 2 demonstrates that the policy induced by the value function trained with Eq.(6) may exhibit the traits of the Lyapunov reliable policy described in Definition 3, fulfilling the reliability criteria in our study.

Then like previous pessimistic methods (An et al., 2021; Bai et al., 2022; Yang et al., 2022), we approximate the uncertainty quantifier in Eq.(4) as the standard deviation as,

$$\Gamma_f(s,a) \approx \beta \cdot Std(f^i(s,a)) = \beta \cdot \sqrt{\frac{1}{K}\sum_{i=1}^{K}\left(f^i(s,a) - \bar{f}(s,a)\right)} \tag{7}$$

where $\{f^i\}_{i=1}^{K}$ is the learned Q-ensembles and $\bar{f}$ is the average of the $K$ Q-esembles, and $\beta$ is the balance-coefficient. Then the objective in Eq.(5) is converted to,

$$\mathcal{L}_{LUC}(s,a,s',a',f) = (1-\beta) \cdot Std(f^i(s,a)) + \beta \cdot \gamma \cdot Std(f^i(s',a')) \tag{8}$$

where $U_f(s,a) = std(f^i(s,a))$ and $\{f^i\}_{i=1}^{K}$ is the learned $K$ Q-ensembles. In practice, the $\beta$ is usually selected in $(0,1)$. Then the regularization of LUC is,

$$\mathcal{L}_{LUC}(f^i,\pi) = \mathbb{E}_{(\hat{s},\hat{a},\hat{s}' \sim \hat{D})}(f^i(\hat{s},\hat{a}) - \hat{\beta} \cdot \mathbb{E}_{\hat{a}' \sim \pi(\cdot|\hat{s}')}\mathcal{L}_{LUC}(\hat{s},\hat{a},\hat{s}',\hat{a}',f)) \tag{9}$$

where $\hat{\mathcal{D}}$ is the constructed noisy dataset. To be specific, the noised samples in $\hat{\mathcal{D}}$, as $\hat{x} = (\hat{s},\hat{a},\hat{s}')$, are the noised version of samples, $x = (s,a,s')$, in the original offline dataset $\mathcal{D}$, with $\hat{x} = x + \lambda \cdot \epsilon$, and $\epsilon$ is the attached perturbation. Previous studies (Bai et al., 2022; Laskey et al., 2017; Zhang et al., 2022; Jiang et al., 2023) have empirically demonstrated the effectiveness of noise injection in regulating the out-of-distribution (OOD) performance of the trained agent. In the majority of our experiments, $\epsilon$ is randomly drawn from a standard Gaussian distribution (also, in the OOD observation experiments in Appendix B.1, $\epsilon$ is generated adversarially as in (Yang et al., 2022)).

Then the loss functions of the ensemble critic networks ($L_c$) and the actor network ($L_a$) are as,

$$\mathcal{L}_c = \mathbb{E}_{(s,a,r,s')\sim\mathcal{D}}\big[\big(r + \gamma\mathbb{E}_{a'\sim\pi(\cdot|s')}[\min_{i=1...K} f'_i(s',a')] - f(s,a)\big)^2 + \frac{1}{K}\sum_{i=1}^{K}\mathcal{L}_{LUC}(f^i,\pi) \quad (10)$$

$$\mathcal{L}_a = \mathbb{E}_{s\sim\mathcal{D}}\mathbb{E}_{a\sim\pi(\cdot|s)}\big[\min_{i=1...K} f'_i(s',a')\big] \quad (11)$$

To sum up, we present our overall approach in Algorithm 1, as follows,

---

**Algorithm 1** The pseudocode of Lyapunov Uncertainty Control (LUC) algorithm

---

**Input**: The offline dataset $\mathcal{D}$. Maximum of episode $T$.

Initialize the policy network, Q-network.

Perform the noise injection to generate the noisy dataset $\hat{\mathcal{D}}$.

**while** $t < T$ **do**

Sample mini-batch of transitions $(s,a,r,s') \sim \mathcal{D}$ and transitions $(\hat{s},\hat{a},\hat{s}') \sim \hat{\mathcal{D}}$

Update the Q-network minimizing $\mathcal{L}_q$ according to Eq.(10)

Update the policy network minimizing $\mathcal{L}_\pi$ according to Eq.(11)

**end while**

**Output:** The learned policy network $\pi$.

---

### 4.3 THEORETICAL ANALYSIS

In this section, by Theorem 2, we aim to show that the value function obtained through k-step iterations of the empirical Bellman operator is determined by two factors concerning the true optimal value function: 1) the single-step Bellman uncertainty generated by policies in the policy candidate set, and 2) the growth tendency of Bellman uncertainty along trajectories generated by policies in the policy candidate set. The former has been the focus of previous methods like PBRL (Bai et al., 2022); however, Theorem 2 in this paper indicates that to learn a better value function, attention must be paid to both factors simultaneously.

**Theorem 2.** *(Performance lower bound.)* *Given an MDP with max reward $R_{max}$ and a dataset of size $N$. Given $(s,a)$ pair, we denote its data density over the dataset is $d(s,a)$. Given an empirical Bellman operator $\hat{\mathcal{T}}^\Pi$ and an arbitrary policy candidate set $\Pi$, where $\hat{\mathcal{T}}^\Pi f(s,a) = r(s,a) + \gamma\mathbb{E}_{s'\sim\hat{P}(s'|s,a)}\max_{\pi\in\Pi} f(s',\pi)$. Denote the learnt value function as $f_k$, with $k$ iterations of $\hat{f}_k = \hat{\mathcal{T}}^\Pi\hat{f}_{k-1}$, and the true optimal value function as $f^*$. Then we have,*

$$\|\hat{f}_k - f^*\|_d \leq \frac{C^*}{1-\gamma}\cdot\sup_{\pi\in\Pi}\sum_{s_0}d(s_0)\zeta_{\hat{f}_k}(s_0,\pi)+$$

$$\mathcal{O}\left(\sup_{\substack{\pi\in\Pi,T\geq 0 \\ s_0,a_0,d(s_0,a_0)>0}}\mathbb{E}_{P(\tau_T|\pi,s_0,a_0)}(\sum_{t=0}^{T-1}[\gamma^{t+1}\zeta_{\hat{f}_k,t+1} - \gamma^t\zeta_{\hat{f}_k,t}])^2\right) \quad (12)$$

*where $\zeta_{\hat{f}_k,t}$ is the Bellman uncertainty at time step $t$, i.e., $\zeta_{\hat{f}_k,t} = \|\mathcal{T}^\Pi\hat{f}_k - \hat{\mathcal{T}}^\Pi\hat{f}_k\|(s_t,a_t)$. $\tau_T$ is the trajectory with length of $T$. And $C^*$ is assumed by $\sup_{s,a}\frac{\pi^*(a|s)}{\pi_\beta(a|s)} \leq C^*$.*

Proof of Theorem 2 is found in Appendix A.2. Theorem 2 primarily shows the following points:

1) The role of LUC term in enhancing the agent's performance - it helps optimize the lower bound of the agent's performance. Specifically, Theorem 2 illustrates that when iterated using the empirical Bellman operator, the difference between the learned value function $\hat{f}_k$ and the true optimal value function $f^*$, which is also known as the performance lower bound of the offline algorithm, can be controlled through the LUC method.

2) One intuitive way to understand Theorem 2 is that we control the right term in Eq.(12) by adjusting the policy candidate set, thereby enhancing the lower bound of the method's output policy

performance. Specifically, we align the policy candidate set with the definition of Lyapunov reliable policy (Definition 3) through the loss function in Eq. (5).

3) Such operation can control the right term of Eq.(12). And then consequently improve the lower bound of the algorithm's performance. The proposed method would not conflict with the objective of approaching the optimal policy, under the assumption of optimal coverage.

Then to further simplify the calculation complexity, Proposition 3 indicates that one-step Lyapunov Uncertainty-penalization could bound the second term in Eq.12.

**Proposition 3.** *If the first term of Eq.(12) is bounded, i.e., $\forall \pi \in \Pi$, we have $\mathbb{E}_{d(s_0)} \zeta_{\hat{f}_k}(s_0, \pi) \leq c$, then we can bound the second term with one-step Lyapunov Uncertainty-penalization, i.e., $\forall s \sim \mathcal{D}$,*

$$\min_{\pi}[\gamma \mathbb{E}_{P(s'|s,\pi)} \zeta_{\hat{f}_k, t+1}(s', \pi) - \zeta_{\hat{f}_k, t+1}(s, \pi)] \Rightarrow \min_{\pi} \mathbb{E}_{P(\tau_T|\pi, s_0=s)} \left( \sum_{t=0}^{T-1} [\gamma^{t+1} \zeta_{\hat{f}_k, t+1} - \gamma^t \zeta_{\hat{f}_k, t}] \right)$$

*Furthermore, assume the dataset fully covers dynamics modes, i.e., $\forall s, a \in \mathcal{D}, supp(P(s'|s, a)) \subseteq supp(\hat{P}(s'|s, a))$, then the left part is controlled by Lyapunov value estimation in Eq.(5).*

Then the left term in Proposition 3 could be empirically estimated by the Lyapunov value estimation as in Definition 3. Theorem 2 and Proposition 3 demonstrate that to control the performance lower bound of the learned agent, despite controlling the current step's Bellman uncertainty, it is also important to control the one-step forward growth tendency of the Bellman uncertainty along with the whole trajectory, which is the main contribution of this paper. This helps the learned value function to be more likely to converge to the fixed point of the empirical Bellman operator, which is hence for controlling the performance lower bound of the learned agent.

## 5 EXPERIMENTAL RESULTS

Our experiments primarily aim to address three key questions:

1. Can LUC enhance the state-of-the-art performance on standard MuJoCo benchmarks?
2. Is LUC capable of consistently learning reliable operation regions from noisy datasets with poor demonstrations?
3. Does LUC exhibit superior generalization ability in avoiding deviations from reliable regions under various types of OOD perturbations?

Our experimental section includes the following components: first, we validate the performance of the method proposed in this paper on standard D4RL benchmarks, particularly on non-expert datasets, demonstrating our method's superiority over others, addressing question 1. Next, to address question 2, we design noise data at different levels - where noise represents the discrepancy between the policy and the optimal policy, resulting in varying degrees of poor demonstrations. We then evaluate the performance of different algorithms on such noisy data. Subsequently, we introduce Out-of-distribution (OOD) MuJoCo benchmarks with various perturbations to increase the likelihood of entering high-uncertainty states, assessing the agent's OOD generalization capability, answering question 3. Finally, we conduct ablation experiments to verify the effectiveness of the LUC method. A brief introduction of our code is provided in Appendix B.2.

### 5.1 LEARNING ON STANDARD MUJOCO BENCHMARKS

We assess our method using the D4RL benchmark (Fu et al., 2020) across various continuous-control tasks and datasets. We compare LUC with several offline RL algorithms, including CQL (Kumar et al., 2020), PBRL (Bai et al., 2022), MOPO (Yu et al., 2020), RORL (Yang et al., 2022), and MOBILE (Sun et al., 2023). Among these, PBRL (Bai et al., 2022), MOPO (Yu et al., 2020), and MOBILE (Sun et al., 2023) are most closely related to LUC as they are all based on uncertainty penalization techniques[1].

---

[1]Unfortunately, as the LDM method (Kang et al., 2022) is mainly used in model-based RL as a constraint on the model optimizer, it is unclear how this method could be properly used for the task of offline RL, so we did not make any comparison with this method at the current stage.

Table 1: Normalized scores on standard MuJoCo tasks, averaged over 4 random seeds. Part of the results are reported in the RORL and MOBILE papers. Top two scores for each task are highlighted. (·) indicates the average without 'expert' datasets.

| | Task Name | CQL | PBRL | MOPO | RORL | MOBILE | LUC (Ours) |
|---|---|---|---|---|---|---|---|
| halfcheetah | random | 31.3±3.5 | 11.0±5.8 | **35.4**±2.5 | 28.5±0.8 | **39.3**±3.0 | 31.3 ± 1.4 |
| | medium | 46.9±0.4 | 57.9±1.5 | 42.3±1.6 | 66.8±0.7 | **74.6**±1.2 | **68.2** ± 1.1 |
| | medium-expert | 95.0±1.4 | 92.3±1.1 | 63.3±38.0 | 107.8±1.1 | 108.2±2.5 | **111.6**± 1.2 |
| | medium-replay | 45.3±0.3 | 45.1±8.0 | 53.1±2.0 | 61.9±1.5 | **71.7**±1.2 | **65.9** ± 1.9 |
| | expert | 97.3±1.1 | 92.4±1.7 | – | 105.2±0.7 | – | **108.3**± 0.5 |
| hopper | random | 5.3±0.6 | 26.8±9.3 | 11.7±0.4 | 31.4±0.1 | **31.9**±0.6 | **31.9** ± 1.4 |
| | medium | 61.9±6.4 | 75.3±31.2 | 28.0±12.4 | 104.8±0.1 | **106.6**±0.6 | **106.9**± 0.4 |
| | medium-expert | 96.9±15.1 | 110.8±0.8 | 23.7±6.0 | **112.7**±0.2 | 112.6±0.2 | **114.3**± 1.1 |
| | medium-replay | 86.3±7.3 | 100.6±1.0 | 67.5±24.7 | 102.8±0.5 | **103.9**±1.0 | **103.6**± 0.7 |
| | expert | 106.5±9.1 | 110.5±0.4 | – | **112.8**±0.2 | – | **114.2**± 0.4 |
| walker2d | random | 5.4±1.7 | 8.1±4.4 | 13.6±2.6 | **21.4**±0.2 | 17.9±3.0 | **25.6** ± 1.2 |
| | medium | 79.5±3.2 | 89.6±0.7 | 17.8±19.3 | **102.4**±1.4 | 87.7±1.1 | **103.6**± 1.3 |
| | medium-expert | 109.1±0.2 | 110.1±0.3 | 44.6±12.9 | **121.2**±1.5 | 115.2±0.7 | **124.1**± 0.9 |
| | medium-replay | 76.8±10.0 | 77.7±14.5 | 39.0±9.6 | **90.4** ±0.5 | 89.9±1.5 | **92.8** ± 1.5 |
| | expert | 109.3±0.1 | 108.3±0.3 | – | **115.4**±0.5 | – | **116.2**± 0.4 |
| Average score | | 70.2 | 74.4 | (36.7) | 85.7 | (80.0) | **87.9 (81.7)** |

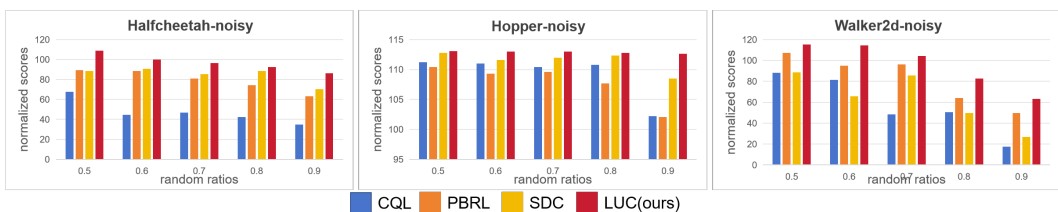

Figure 2: Results of CQL, PBRL SDC and LUC on tasks with different levels of non-expert data.

The results are presented in Table 1. It is evident that our method, LUC, outperforms other methods in most tasks and achieves a higher overall score. Particularly, LUC performs significantly better on non-expert datasets, notably the 'medium-expert' dataset, showcasing its robustness against the reward shift due to the conservative regularization term on non-expert data. Additionally, LUC's performance in the 'hopper' and 'walker2d' environments surpasses the State-of-the-art (SOTA), possibly due to these environments being more susceptible to noise from non-expert data.

## 5.2 LEARNING ON TASKS WITH DIFFERENT LEVELS OF NON-EXPERT NOISE

In this section, we modify the discrepancy between the behavioral policy and the optimal policy by blending datasets produced from expert and random policies at various proportions to generate noisy datasets at different levels. Evaluating these datasets not only validates the influence of non-expert problems on conventional conservative constraints but also confirms the robustness of the proposed LUC method against noise stemming from non-expert data. We compare several representative methods: CQL (Kumar et al., 2020), PBRL (Bai et al., 2022), and SDC (Zhang et al., 2022).

The results are depicted in Figure 2. It is evident that most constraint-based methods are influenced by non-expert issues, wherein an increase in the discrepancy between the behavioral policy and the optimal policy leads to a substantial performance decline. However, as the randomness level rises, the performance degradation of LUC is comparatively minor, suggesting that LUC demonstrates greater resilience to noise from non-expert data under suboptimal behavioral policy settings. Particularly at elevated randomness levels (e.g., 0.9), LUC can maintain effective performance on these benchmarks.

## 5.3 TESTING ON OUT-OF-DISTRIBUTION MUJOCO BENCHMARKS

To evaluate the agent's capacity to avoid straying from reliable regions, we introduce three types of OOD perturbations, applying varying intensities of perturbations at different intervals to the agent

employed for a higher risk of deviation. This investigation encompasses three combinations of noise intensities and intervals[2]. It is important to highlight that our approach in this study differs from the perturbation noise discussed in Yang et al. (2022); our method modifies the actual state in which the agent operates, rather than solely perturbing state observations. The research is centered on three MuJoCo environments: Halfcheetah, Hopper, and Walker2d; with the agent trained on 'medium-expert' datasets.

Table 2: The results of OSR, RORL and LUC (ours) on Out-of-distribution MuJoCo benchmarks. The highest scores for each task are highlighted.

|  | Halfcheetah-ood | | | Hopper-ood | | | Walker2d-ood | | |
|---|---|---|---|---|---|---|---|---|---|
|  | small | medium | large | small | medium | large | small | medium | large |
| OSR | 93.8 | 90.4 | 88.7 | 111.3 | 103.4 | **85.7** | 112.7 | 110.5 | 105.8 |
| RORL | 102.7 | 94.3 | 82.4 | **111.5** | 92.8 | 72.7 | 117.4 | 107.3 | 86.9 |
| LUC(ours) | **104.8** | **102.9** | **99.0** | 110.9 | **106.6** | 83.3 | **119.4** | **114.5** | **111.9** |

We have chosen two key algorithms, OSR and RORL, tailored for managing OOD states and observations, to contrast with the proposed LUC in these OOD benchmarks. The outcomes are detailed in Table 2. Analysis reveals that LUC surpasses the other two methods across the majority of tasks, notably demonstrating substantial benefits in extensive OOD perturbation assignments like Halfcheetah and Walker2d. This implies that these settings might be more sensitive to OOD perturbations, necessitating advanced the agent to tackle OOD scenarios.

To delve deeper into the factors contributing to the superior performance of LUC in Halfcheetah-ood and Walker2d-ood tasks, we present the visualized results of LUC in Figure 3. The analysis reveals that each perturbation event leads the agent into high-uncertainty regions; however, LUC effectively prevents error accumulation and guides the agent back to lower-error (reliable) regions. This underscores the robustness of LUC in handling OOD scenarios by constraining the agent to operate within the reliable regions.

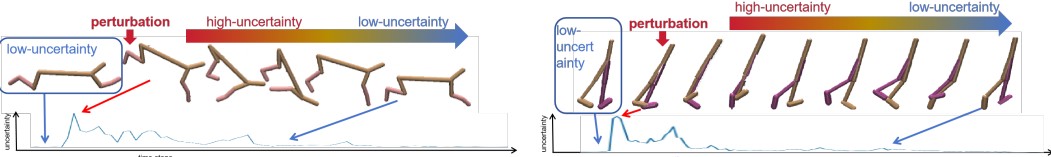

Figure 3: Visualized results of LUC on OOD MuJoCo benchmarks - 'Halfcheetah' and 'Walker2d', with large scales of perturbations. The Bellman uncertainty (error) is estimated by the standard deviation uncertainty based on the learned Q-ensembles. The interval between every two frames is about five steps.

## 5.4 MORE COMPLICATED ENVIRONMENTS - ANTMAZE AND ADROIT

Compared to the MuJoCo environment, the AntMaze and Adroit environments require the agent to have the ability of multi-step dynamic planning, making them considered a more complex scenario. The results are shown in Table 3 and 4. In the AntMaze environment, based on the size and shape of the maze, it can be categorized into 'umaze,' 'medium,' and 'large'; and based on different tasks, it can be classified as 'diverse' and 'play'. While the Adroit domain features three types of datasets: demonstration data from humans ("human"), expert data from a reinforcement learning policy ("expert"), and mixed data combining human demonstrations with an imitation policy ("cloned"). The tasks in Adroit are more complex than those in the Gym domain, and the inclusion of human demonstrations adds an additional layer of difficulty.

---

[2]Details regarding the construction of Out-of-distribution MuJoCo benchmarks are outlined in Appendix B.3.

Here, we compare CQL (Kumar et al., 2020), IQL (Kostrikov et al., 2022), SPOT (Wu et al., 2022), ATAC (Cheng et al., 2022), SDC (Zhang et al., 2022), and OSR-10 (Jiang et al., 2023) in AntMaze, while CQL (Kumar et al., 2020) and PBRL (Bai et al., 2022) in Adroid.

Table 3: Results of **LUC(ours)**, CQL, IQL, SPOT, ATAC, SDC and OSR-10 on offline AntMaze tasks averaged over 4 seeds. We bold the highest scores in each task.

|  |  | CQL | IQL | SPOT | ATAC | SDC | OSR-10 | **LUC(Ours)** |
|---|---|---|---|---|---|---|---|---|
|  | umaze | 82.6 | 87.5 | 93.5 | 70.6 | 89.0 | 89.9 | **94.3±1.3** |
|  | umaze-diverse | 10.2 | 62.2 | 40.7 | 54.3 | 57.3 | **74.0** | 62.1±3.7 |
| AntMaze | medium-play | 59.0 | 71.2 | 74.7 | 72.3 | 71.9 | 66.0 | **80.1±2.2** |
|  | medium-diverse | 46.6 | 70.0 | 79.1 | 68.7 | 78.7 | **80.0** | 78.7±3.9 |
|  | large-play | 16.4 | 39.6 | 35.3 | 38.5 | 37.2 | 37.9 | **44.3±7.2** |
|  | large-diverse | 3.2 | **47.5** | 36.3 | 43.1 | 33.2 | 37.9 | 41.7±6.1 |
| average |  | 36.3 | 63.0 | 59.9 | 57.9 | 61.2 | 64.3 | **66.9** |

Table 4: Results of **LUC(ours)**, CQL and PBRL on offline Adroit tasks averaged over 4 seeds. We bold the highest scores in each task.

|  | pen-hu. | pen-cl. | pen-ex. | hammer-hu. | hammer-cl. | hammer-ex. | door-hu. | door-cl. | door-ex. | avg. |
|---|---|---|---|---|---|---|---|---|---|---|
| CQL | 37.5 | 39.2 | 107.0 | 4.4 | 2.1 | 86.7 | 9.9 | 0.4 | 101.5 | 43.2 |
| PBRL | 35.4 | **74.9** | 137.7 | 0.4 | 0.8 | 127.5 | 0.1 | 4.6 | 95.7 | 53.0 |
| **LUC(ours)** | **41.6** | 68.2 | **142.5** | **7.9** | **4.7** | **130.4** | 7.8 | **8.4** | **103.4** | **57.2** |

From these benchmarks, we observe that our method outperforms the other methods in most benchmarks. In particular, our method performs much better in the "hammer," "door," and "large" maze benchmarks than other methods. This could be attributed to our method's ability to scope a safe region for the agent to stably operate within, thus addressing more complex environmental dynamics.

### 5.5 ABLATION STUDY

In this section, we performed ablation experiments on the proposed LUC method to confirm its contribution to the overall framework. The findings, illustrated in Figure 4, demonstrate a substantial performance enhancement in the LUC module compared to LUC without reward shaping, particularly on certain non-expert datasets. This highlights the pivotal role of the LUC method in improving the performance of pessimistic offline RL approaches.

*Furthermore, we conducted experiments on Out-of-Distribution (OOD) observation benchmarks to further validate the robustness of the proposed method, as detailed in Appendix B.1. More in-* *formation on additional experiments, including parameters and more, can be found in Appendix B.*

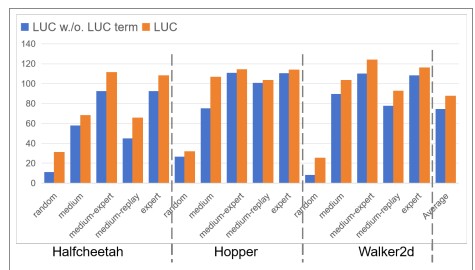

Figure 4: Ablation study.

## 6 CONCLUSION

This paper aims to identify a reliable operational region for the agent based on offline data. To achieve this, we introduce the Lyapunov Uncertainty Control (LUC) algorithm in an offline, model-free manner. Theoretically, in deterministic MDPs or when the dataset fully covers all dynamic modes, LUC can confine the agent's operations within low-uncertainty areas, thereby enhancing decision-making reliability. Empirically, LUC-trained agents demonstrate superior robustness and reliability in high-risk scenarios compared to various other methods. In future works, LUC can serve as a versatile tool in diverse offline reinforcement learning frameworks, including model-based approaches, potentially paving the way for new research opportunities.

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

## A   APPENDIX

### A.1   PROOFS OF MAIN THEOREMS IN SEC.4.1

**Proposition 1. (Existence of reliable policy.)** Suppose the dataset have a sufficient coverage over the optimal policy, i.e., $\sup_{s,a} \frac{\pi^*(a|s)}{\pi_\beta(a|s)} \leq C^*$). Then there exists a reliable policy.

*Proof sketch.*   From the assumption that the dataset have a sufficient over the optimal policy, we have all the behaviors of the optimal policy would be supported by the dataset, i.e., $\pi^* \subseteq \mathcal{D}$. Then the optimal stationary state distribution is also supported by the dataset $d^{\pi^*}(s) \subseteq \mathcal{D}$. Due to the fact that all the states over the dataset are contained in the $f - \pi^*$ reliable region, and $d^{\pi^*}(s) \subseteq \mathcal{D}$, so the optimal policy would never operate beyond its $f - \pi^*$ reliable region, hence it is a reliable policy.

**Theorem 1.** In an MDP with deterministic transition, Lyapunov policy $\pi$ is a reliable policy as defined in Definition 2.

*Proof of Theorem 1.*   We prove this theorem in a contradiction way. Denote the deterministic transition as $M(s,a) = s'$. First, we suppose the Lyapunov policy $\pi$ is not a reliable policy, then we have: $\forall s_0 \in \mathcal{D} \cap \mathcal{G}_f(\pi), \exists t, s_t \in supp(P(s_t|s_0,\pi))$, such that $s_t \notin \mathcal{G}_f(\pi)$. Then we have $\zeta_f(s_t,\pi) > c$. Then we aim to find the contradiction. From the definition of Lyapunov policy, $\forall s \in \mathcal{D}$, we have,

$$c < \zeta_f(s_t,\pi) \leq \max_{\hat{a}\in\pi} \zeta_f(P(s_{t-1},\hat{a}),\pi) \leq \epsilon_f(s_{t-1},\pi) \tag{13}$$

Then we have $\epsilon_f(s_{t-1},\pi) > c$. By recurrently applying the above derivation by $t$ times, we would have $\zeta_f(s_0,\pi) > c$. Then we have $s_0 \notin \mathcal{G}_f(\pi)$, which is conflicted with $s_0 \in \mathcal{D}\cap\mathcal{G}_f(\pi)$. Completing the proof, and we can conclude that the Lyapunov policy is reliable.

**Proposition 2.** Suppose the action distribution of new policy $\pi(a|s)$ is positive correlated to the learnt Q function $f(s,a)$, i.e., $\pi(a|s) \propto f(s,a)$. Then the proposed Lyapunov value estimation induces a Lyapunov policy as defined in Definition 3.

*Proof of Proposition 2.*   Denote the empirical behavior of the dataset $\mathcal{D}$ as $\pi_\beta$, whose actions are always supported by the dataset. The Lyapunov value estimation implicitly penalize the new policy $\pi$ at an given state $s$ with two aspects:

1) $\min_\pi \mathbb{E}_{a\sim\pi(a|s)}\zeta_f(s,a)$; This constrains the new policy would not generated OOD actions beyond the demonstration of the offline data, i.e., $supp(\pi(a|s)) \subset supp(\pi_\beta(a|s))$. In previous works Wu et al. (2022); Mao et al. (2024), such supported constraint is often achieved in a data density based way as $\min_\pi \sum_{a\notin\pi_\beta} \pi(a|s)$. Then we will show that the current step's error controlling minimizes the upper bound of the above supported constraint,

$$\min_\pi \sum_{a\notin\pi_\beta} \pi(a|s) \Leftrightarrow \max_\pi \mathbb{E}_{a\in\pi(a|s)}d(s,a) \tag{14}$$

$$\leq^{(a)} \max_\pi C \cdot \mathbb{E}_{a\in\pi(a|s)} \frac{1}{\zeta_f^2(s,a)} \tag{15}$$

$$\Leftrightarrow \min_\pi \mathbb{E}_{a\in\pi(a|s)}\zeta_f^2(s,a) \tag{16}$$

$$\Leftrightarrow^{(b)} \min_\pi \mathbb{E}_{a\in\pi(a|s)}\zeta_f(s,a) \tag{17}$$

The inequality $(a)$ holds because of the Lemma 1 in Appendix A.2. The equivalence $(b)$ holds because the Bellman uncertainty is always non-negative.

Then with this constraint, we can have the new policy would reject the actions that is not supported by the behavior policy. This means, the new policy would only select the data-supported actions.

2) $\forall s,a \in \mathcal{D}$, we have $\zeta_f(M(s,a),\pi) \leq \zeta_f(s,a)$. This could also be consider as,

$$\min_{a\in\pi_\beta} \zeta_f(s,a) - \zeta_f(M(s,a),\pi) \tag{18}$$

$$\leq^{(a)} \min_{a\in\pi} \zeta_f(s,a) - \zeta_f(M(s,a),\pi) \tag{19}$$

$$\leq \zeta_f(s,a_m) - \zeta_f(M(s,a_m),\pi) \tag{20}$$

where $a_m = \arg\max_{a_m \in \pi} \zeta_f(M(s, a_m), \pi)$. The $(a)$ holds because of the assumption that the condition 1) is perfectly confirmed, i.e., $supp(\pi(a|s)) \subset supp(\pi_\beta(a|s))$.

Then we have $\forall s \in \mathcal{D}, \max_{\hat{a} \in \pi} \zeta_f(M(s, \hat{a}), \pi) \leq \zeta_f(s, \pi)$. And we can conclude that the Lyapunov value estimation would help to induce a Lyapunov policy.

## A.2 Proofs of Main Theorem with Stochastic Transition Setting

First we define the policy candidate set $\Pi$ based on a version space of all the functions $f \in \mathcal{F}$, where we have $\forall \pi \in \Pi, \exists f \in \mathcal{F}, \pi(a|s) \propto f(s, a)$, and $\forall f \in \mathcal{F}, \exists \pi \in \Pi, \pi$ is the greedy policy according to $f$. Then we define a corresponding Bellman operator $\mathcal{T}^\Pi f(s, a) = r + \gamma \mathbb{E}_{s' \sim P} \max_{\pi \in \Pi} f(s', \pi)$ and its empirical version is $\hat{\mathcal{T}}^\Pi f(s, a) = r + \gamma \mathbb{E}_{s' \sim \hat{P}} \max_{\pi \in \Pi} f(s', \pi)$. Then before the introduction of theoretical results, a basic assumption should be made.

**Assumption 1.** *(Optimal coverage.) (Xie et al., 2021) We assume the dataset have sufficient coverage over the optimal policy's visitation, i.e., $\sup_{s,a} \frac{\pi^*(a|s)}{\pi_\beta(a|s)} \leq C^*$, and $\pi^* \in \Pi$.*

Similar assumptions has been utilized in theoretical analysis for offline RL (Xie et al., 2021). Compared with the more common assumption - Concentrability assumption (Munos, 2005; Kumar et al., 2019) that the dataset should fully cover the whole state space, Assumption 1 is much looser and more feasible in practice.

**Definition 4.** *(Recoverability) Define the recoverability of a given policy $\pi$ from the given $(s_0, a_0)$ pair,*

$$R(\pi)\big|_{s_0,a_0} = \inf_{T \geq 0} \mathbb{E}_{s_T, a_T \sim P(s_T, a_T | \pi, s_0, a_0)} \frac{d(s_T, a_T)}{d(s_0, a_0)} \tag{21}$$

*where $d(s, a)$ is the data density at $(s, a)$.*

**Definition 5.** *(Recoverability risk) We define the most risk of a policy $\pi$ that the agent is able to recover to the regions with low Bellman uncertainty, i.e., its familiar regions, from the given $(s_0, a_0)$ pair,*

$$Risk(\pi)\big|_{s_0,a_0} = \sup_{T \geq 0} \mathbb{E}_{s_T, a_T \sim P(s_T, a_T | \pi, s_0, a_0)} \frac{\|\mathcal{T}^\pi f - \hat{\mathcal{T}}^\pi f\|(s_T, a_T)}{\|\mathcal{T}^\pi f - \hat{\mathcal{T}}^\pi f\|(s_0, a_0)} = \sup_{T \geq 0} \mathbb{E}_{s_T, a_T} \frac{\zeta_T}{\zeta_0} \tag{22}$$

*where $\mathcal{T}^\pi$ is the true Bellman operator and $\hat{\mathcal{T}}^\pi$ is the empirical Bellman.*

**Lemma 1.** *Given an MDP with max reward $R_{max}$ and a dataset of size $N$. The dimension of state space is $|S|$ and that of action space is $|A|$. Given $(s, a)$ pair, we denote its data density over the dataset is $d(s, a)$. Then with probability $1 - \delta$, we have,*

$$d(s, a) \leq \gamma^2 \cdot R_{max}^2 \frac{2}{N \cdot \|\mathcal{T}^\pi f - \hat{\mathcal{T}}^\pi f\|^2(s, a)} \log(\frac{|S||A| \cdot 2^{|S|}}{\delta}) \tag{23}$$

*where $\|\mathcal{T}^\pi f - \hat{\mathcal{T}}^\pi f\|(s, a)$ is Bellman uncertainty.*

*Proof of Lemma 1.*

$$\|\mathcal{T}^\pi f - \hat{\mathcal{T}}^\pi f\|(s, a) = \gamma\|\sum_{s'}(\hat{P}(s'|s, a) - P(s'|s, a)) \cdot f(s', \pi)\| \tag{24}$$

$$\leq \gamma\|\sum_{s'}(\hat{P}(s'|s, a) - P(s'|s, a))\| \cdot \|f(s', \pi)\| \tag{25}$$

$$\leq \gamma \cdot R_{max} \cdot \|\hat{P}(s'|s, a) - P(s'|s, a)\|_1 \tag{26}$$

$$\leq^{(a)} \gamma \cdot R_{max} \cdot \sqrt{\frac{2}{N(s, a)} \log(\frac{|S||A| \cdot 2^{|S|}}{\delta})} \tag{27}$$

$$\Rightarrow d(s, a) \leq \gamma^2 \cdot R_{max}^2 \frac{2}{N \cdot \|\mathcal{T}^\pi f - \hat{\mathcal{T}}^\pi f\|^2(s, a)} \log(\frac{|S||A| \cdot 2^{|S|}}{\delta}) \tag{28}$$

*The inequality $(a)$ holds because of the **Proposition 9** in Ghavamzadeh et al. (2016). And $N(s,a)$ is the number of $(s,a)$ samples in the dataset, so the density $d(s,a) = \frac{N(s,a)}{N}$. Completing the proof.*

**Then we give Lemma 2, which is a general formulation of Lemma ?? in the main text.**

**Lemma 2.** *For any policy $\pi$ and $(s_0, a_0)$ pair, we have,*

$$Risk(\pi)\big|_{s_0,a_0} = \sup_{T \geq 0} \mathbb{E}_\pi \left[ \frac{\sum_{t=0}^{T-1}[\gamma^{t+1}\zeta_{t+1} - \gamma^t\zeta_t] + \zeta_0}{\zeta_0 \cdot \gamma^T}\Big|s_0, a_0 \right] \tag{29}$$

*Proof of Lemma 2.*

$$Risk(\pi)\big|_{s_0,a_0} = \sup_{T \geq 0} \mathbb{E}_\pi \left[ \frac{\zeta_T}{\zeta_0}\big|s_0, a_0 \right] \tag{30}$$

$$= \sup_{T \geq 0} \mathbb{E}_\pi \left[ \frac{\gamma^T\zeta_T - \gamma^{T-1}\zeta_{T-1} + \gamma^{T-1}\zeta_{T-1} - ... + \gamma\zeta_1 - \zeta_0 + \zeta_0}{\zeta_0 \cdot \gamma^T}\big|s_0, a_0 \right] \tag{31}$$

$$= \sup_{T \geq 0} \mathbb{E}_\pi \left[ \frac{\sum_{t=0}^{T-1}[\gamma^{t+1}\zeta_{t+1} - \gamma^t\zeta_t] + \zeta_0}{\zeta_0 \cdot \gamma^T}\big|s_0, a_0 \right] \tag{32}$$

Completing the proof.

**Lemma 3.** *Given an arbitrary Bellman operator (maybe empirical Bellman operator) $\mathcal{T}$ and an arbitrary policy candidate set $\Pi$. We have $\mathcal{T}f(s,a) = r + \gamma\mathbb{E}_{s'}\max_{\pi\in\Pi} f(s',\pi)$. Then for any value function $f_1, f_2 \in \mathcal{F}$ and $t \geq 0$, we have,*

$$\|\mathcal{T}^{(t)}f_1 - \mathcal{T}^{(t)}f_2\|_d \leq \sup_{s_0,a_0,d(s_0,a_0)>0} \frac{\gamma^t}{R(\hat{\pi})\big|_{s_0,a_0}}\|f_1 - f_2\|_d \tag{33}$$

*We denote the greedy policy induced by $f_1$ as $\pi_1$ and the greedy policy induced by $f_2$ as $\pi_2$. Then $\hat{\pi}$ is the pessimistic policy of $f_1$ and $f_2$, i.e., $\hat{\pi}(a|s) = \pi_1(a|s)$ if $f_1(s,\pi_1) \leq f_2(s,\pi_2)$ and $\hat{\pi}(a|s) = \pi_2(a|s)$ if $f_2(s,\pi_2) \leq f_1(s,\pi_1)$. And $\|x\|_d = \sum_x d(x)|x|$ is a distributional weighted norm, where $d$ here is the density of the dataset.*

*Proof of Lemma 3.* First we denote $\pi_1(a|s) = \arg\max_{\pi\in\Pi} f_1(s,\pi)$ and $\pi_2(a|s) = \arg\max_{\pi\in\Pi} f_2(s,\pi)$. Then,

$$(\mathcal{T}f_1 - \mathcal{T}f_2)(s,a) = \gamma\mathbb{E}_{P(s'|s,a)}[f_1(s',\pi_1) - f_2(s',\pi_2)] \tag{34}$$

$$\leq \gamma\mathbb{E}_{P(s'|s,a)}[f_1(s',\pi_1) - f_2(s',\pi_1)] \tag{35}$$

On the other hand,

$$(\mathcal{T}f_1 - \mathcal{T}f_2)(s,a) \geq \gamma\mathbb{E}_{P(s'|s,a)}[f_1(s',\pi_2) - f_2(s',\pi_2)] \tag{36}$$

Then we construct $\hat{\pi}(a|s) = \pi_1(a|s)$ if $f_1(s,\pi_1) \leq f_2(s,\pi_2)$ and $\hat{\pi}(a|s) = \pi_2(a|s)$ if $f_2(s,\pi_2) \leq f_1(s,\pi_1)$. So we have,

$$|\mathcal{T}f_1 - \mathcal{T}f_2|(s,a) \leq \gamma|\mathbb{E}_{P(s'|s,a)}[f_1(s',\hat{\pi}) - f_2(s',\hat{\pi})]| \tag{37}$$

Then if we recursively apply the $\hat{\pi}$, we would have,

$$|\mathcal{T}^{(t)}f_1 - \mathcal{T}^{(t)}f_2|(s,a) \leq \gamma^t \cdot |\mathbb{E}_{P(s_t,a_t|\hat{\pi},s_0=s,a_0=a)}[f_1(s_t,a_t) - f_2(s_t,a_t)]| \tag{38}$$

Then we aim to bound the $\|\mathcal{T}^{(t)}f_1 - \mathcal{T}^{(t)}f_2\|_d$,

$$\|\mathcal{T}^{(t)}f_1 - \mathcal{T}^{(t)}f_2\|_d = \sum_{s_0,a_0} |\mathcal{T}^{(t)}f_1 - \mathcal{T}^{(t)}f_2|(s_0,a_0)d(s_0,a_0) \tag{39}$$

$$\leq \gamma^t \cdot \sum_{s_0,a_0} |\mathbb{E}_{P(s_t,a_t|\hat{\pi},s_0,a_0)}[f_1(s_t,a_t) - f_2(s_t,a_t)]|d(s_0,a_0) \tag{40}$$

$$\leq \gamma^t \cdot \sum_{s_0,a_0} \sum_{s_t,a_t} d(s_0,a_0)P(s_t,a_t|\hat{\pi},s_0,a_0)|[f_1(s_t,a_t) - f_2(s_t,a_t)]| \tag{41}$$

$$\leq \sup_{s_0,a_0,d(s_0,a_0)>0} \frac{\gamma^t}{R(\hat{\pi})\big|_{s_0,a_0}} \cdot \sum_{s_t,a_t} d(s_t,a_t)|[f_1(s_t,a_t) - f_2(s_t,a_t)]| \tag{42}$$

$$= \sup_{s_0,a_0,d(s_0,a_0)>0} \frac{\gamma^t}{R(\hat{\pi})\big|_{s_0,a_0}} \cdot \|[f_1(s_t,a_t) - f_2(s_t,a_t)]\|_d \tag{43}$$

Completing the proof.

**Corollary 1.** *Especially, if the $f_2$ in Lemma 3 is the fix point of $\mathcal{T}$, as $f^*$, then we have,*

$$\|\mathcal{T}^{(t)}f_1 - f^*\|_d \leq \sum_{s_0,a_0} \frac{\gamma^t}{R(\pi_1)\big|_{s_0,a_0}} \|f_1 - f^*\|_d \tag{44}$$

This is easily obtained by the fact that $\forall s, f^*(s, \pi^*) \geq f_1(s, \pi_1)$.

**Lemma 4.** *Denote the learnt value function as $f_k$, with $k$ iterations of $f_k = \mathcal{T}f_{k-1}$, and the fixed point of $\mathcal{T}$ is $f^*$. Then,*

$$\|f_k - f^*\|_d \leq R(\pi_0) \cdot \gamma^k \cdot \|\triangle_{(0)}\|_d + \epsilon_{max} \cdot \sum_{t=1}^{k}[R(\hat{\pi}_{t-1}) \cdot \gamma^t] + \epsilon_{max} \tag{45}$$

*where $R(\pi_k) = \sum_{s_0,a_0} \frac{1}{R(\pi_k)\big|_{s_0,a_0}}$, $\epsilon_{max} = \max_{t \leq k-1} \|f_{t+1} - \mathcal{T}f_t\|_d$ and $\triangle_{(0)} = \|f_0 - f^*\|_\infty$.*

*Proof of Lemma 4.*

$$\|f_k - f^*\|_d \leq \|\mathcal{T}f_{k-1} - f^*\|_d + \|f_k - \mathcal{T}f_{k-1}\|_d \tag{46}$$

$$\leq \|\mathcal{T}^{(2)}f_{k-2} - f^*\|_d + \|\mathcal{T}f_{k-1} - \mathcal{T}^{(2)}f_{k-2}\|_d + \epsilon_{max} \tag{47}$$

$$\leq \|\mathcal{T}^{(2)}f_{k-2} - f^*\|_d + R(\pi_{k-1}) \cdot \gamma \cdot \epsilon_{max} + \epsilon_{max} \tag{48}$$

$$\ldots\ldots\ldots\ldots \tag{49}$$

$$\leq \|\mathcal{T}^{(k)}f_0 - f^*\|_d + \epsilon_{max} \cdot \sum_{t=1}^{k}[R(\hat{\pi}_{t-1}) \cdot \gamma^t] + \epsilon_{max} \tag{50}$$

$$\leq^{(a)} R(\pi_0) \cdot \gamma^k \cdot \|\triangle_{(0)}\|_d + \epsilon_{max} \cdot \sum_{t=1}^{k}[R(\hat{\pi}_{t-1}) \cdot \gamma^t] + \epsilon_{max} \tag{51}$$

The inequality (a) holds because of Corollary 1. Completing the proof.

**Lemma 5.** *Given an MDP with max reward $R_{max}$ and a dataset of size $N$. The dimension of state space is $|S|$ and that of action space is $|A|$. Given $(s, a)$ pair, we denote its data density over the dataset is $d(s, a)$. Given an empirical Bellman operator $\hat{\mathcal{T}}$ and an arbitrary policy candidate set $\Pi$, where $\hat{\mathcal{T}}f(s, a) = r(s, a) + \gamma \mathbb{E}_{s' \sim \hat{P}(s'|s,a)} \max_{\pi \in \Pi} f(s', \pi)$. Denote the learnt value function as $f_k$, with $k$ iterations of $\hat{f}_k = \hat{\mathcal{T}}\hat{f}_{k-1}$, and the fixed point of $\hat{\mathcal{T}}$ is $\hat{f}^*$. Then we have,*

$$\|\hat{f}_k - \hat{f}^*\|_d \leq \mathcal{O}\left(\sup_{s_0,a_0,d(s_0,a_0)>0} \sup_{\pi \in \Pi, T \geq 0} \mathbb{E}_{s_T,a_T \sim P(s_T,a_T|\pi,s_0,a_0)}(\sum_{t=0}^{T-1} \gamma^{t+1}\zeta_{t+1} - \gamma^t\zeta_t)^2\right) \tag{52}$$

*where $\zeta_t$ is the Bellman uncertainty at time step $t$, i.e., $\zeta_t = \|\mathcal{T}^\pi f - \hat{\mathcal{T}}^\pi f\|(s_t, a_t)$.*

*Proof of Lemma 5.* From Lemma 4 we have known that if we want to bound $\|\hat{f}_k - \hat{f}^*\|_d$, we should bound $\frac{1}{R(\pi)\big|_{s_0,a_0}}$ at each time steps.

$$\frac{1}{R(\pi)\big|_{s_0,a_0}} = \sup_{T \geq 0} \mathbb{E}_{s_T,a_T \sim P(s_T,a_T|\pi,s_0,a_0)} \frac{d(s_0,a_0)}{d(s_T,a_T)} \tag{53}$$

With Lemma 1, we have,

$$\frac{1}{R(\pi)\big|_{s_0,a_0}} = \sup_{T \geq 0} \mathbb{E}_{s_T,a_T \sim P(s_T,a_T|\pi,s_0,a_0)} \frac{d(s_0,a_0)}{d(s_T,a_T)} \tag{54}$$

$$\leq^{(a)} \sup_{T \geq 0} \left(\mathbb{E}_{s_T,a_T \sim P(s_T,a_T|\pi,s_0,a_0)} \frac{\|\mathcal{T}^\pi f - \hat{\mathcal{T}}^\pi f\|(s_T,a_T)}{\|\mathcal{T}^\pi f - \hat{\mathcal{T}}^\pi f\|(s_0,a_0)}\right)^2 \tag{55}$$

The inequality (a) holds because of $d(s_0, a_0) \cdot \zeta_0 \leq \sup_T \mathbb{E}_{s_T, a_T} d(s_T, a_T) \cdot \zeta_T$. Then following Lemma **??**, we have,

$$\sup_{T \geq 0} \mathbb{E}_{s_T, a_T \sim P(s_T, a_T | \pi, s_0, a_0)} \frac{\|\mathcal{T}^\pi f - \hat{\mathcal{T}}^\pi f\|^2 (s_T, a_T)}{\|\mathcal{T}^\pi f - \hat{\mathcal{T}}^\pi f\|^2 (s_0, a_0)} \tag{56}$$

$$= \sup_{T \geq 0} \mathbb{E}_{s_T, a_T \sim P(s_T, a_T | \pi, s_0, a_0)} \left( \frac{\sum_{t=0}^{T-1} [\gamma^{t+1} \zeta_{t+1} - \gamma^t \zeta_t]}{\zeta_0 \cdot \gamma^T} + \frac{1}{\gamma^T} \right)^2 \tag{57}$$

$$\leq \mathcal{O} \left( \sup_{\pi \in \Pi, T \geq 0} \mathbb{E}_{s_T, a_T \sim P(s_T, a_T | \pi, s_0, a_0)} (\sum_{t=0}^{T-1} \gamma^{t+1} \zeta_{t+1} - \gamma^t \zeta_t)^2 \right) \tag{58}$$

Completing the proof.

Please note that Lemma 5 holds for any estimation value function $f$. And we can utilize the learned $\hat{f}_k$ at the $k^{th}$ iteration.

**Furthermore, by plugging Eq.(58) in Corollary 1, we would have the proof for Lemma ??.** Then we give the proof for Theorem 2 in the main text.

**Theorem 2.** *Given an MDP with max reward $R_{max}$ and a dataset of size $N$. The dimension of state space is $|S|$ and that of action space is $|A|$. Given $(s, a)$ pair, we denote its data density over the dataset is $d(s, a)$. Given an empirical Bellman operator $\hat{\mathcal{T}}^\Pi$ and an arbitrary policy candidate set $\Pi$, where $\hat{\mathcal{T}}^\Pi f(s, a) = r(s, a) + \gamma \mathbb{E}_{s' \sim \hat{P}(s' | s, a)} \max_{\pi \in \Pi} f(s', \pi)$. Denote the learnt value function as $f_k$, with $k$ iterations of $\hat{f}_k = \hat{\mathcal{T}}^\Pi \hat{f}_{k-1}$, and the true optimal value function as $f^*$. Then we have,*

$$\|\hat{f}_k - f^*\|_d \leq \frac{C^*}{1 - \gamma} \cdot \sup_{\pi \in \Pi} \sum_{s_0} d(s_0) \zeta(s_0, \pi) + \tag{59}$$

$$\mathcal{O} \left( \sup_{\substack{\pi \in \Pi, T \geq 0 \\ s_0, a_0, d(s_0, a_0) > 0}} \mathbb{E}_{s_T, a_T \sim P(s_T, a_T | \pi, s_0, a_0)} (\sum_{t=0}^{T-1} \gamma^{t+1} \zeta_{t+1} - \gamma^t \zeta_t)^2 \right) \tag{60}$$

*where $\zeta_t$ is the Bellman uncertainty at time step $t$, i.e., $\zeta_t = \|\mathcal{T}^\Pi \hat{f}_k - \hat{\mathcal{T}}^\Pi \hat{f}_k\| (s_t, a_t)$.*

*Proof of Theorem 2.*

$$\|\hat{f}_k - f^*\|_d \leq \|\hat{f}_k - \hat{f}^*\|_d + \|\hat{f}^* - f^*\|_d \tag{61}$$

where $\hat{f}^*$ is the fixed point of $\hat{\mathcal{T}}^\Pi$. Then Lemma 5 bounds $\|\hat{f}_k - \hat{f}^*\|_d$, i.e.,

$$\|\hat{f}_k - \hat{f}^*\|_d \leq \mathcal{O} \left( \sup_{\substack{\pi \in \Pi, T \geq 0 \\ s_0, a_0, d(s_0, a_0) > 0}} \mathbb{E}_{s_T, a_T \sim P(s_T, a_T | \pi, s_0, a_0)} (\sum_{t=0}^{T-1} \gamma^{t+1} \zeta_{t+1} - \gamma^t \zeta_t)^2 \right) \tag{62}$$

On the other hand,

$$\|\hat{f}^* - f^*\|_d \leq \|\hat{\mathcal{T}}^\Pi \hat{f}^* - \hat{\mathcal{T}}^\Pi f^*\|_d + \|\hat{\mathcal{T}}^\Pi f^* - \mathcal{T}^\Pi f^*\|_d \tag{63}$$

$$\leq \|\hat{\mathcal{T}}^\Pi f^* - \mathcal{T}^\Pi f^*\|_d + \gamma \|\hat{f}^* - f^*\|_d \tag{64}$$

$$\Rightarrow \|\hat{f}^* - f^*\|_d \leq \frac{\|\hat{\mathcal{T}}^\Pi f^* - \mathcal{T}^\Pi f^*\|_d}{1 - \gamma} \tag{65}$$

Then due to the optimal coverage assumption that $\sup_{s,a} \frac{\pi^*(a|s)}{\pi_\beta(a|s)} \leq C^*$, and $\pi^* \in \Pi$, we have,

$$\|\hat{\mathcal{T}}^{\Pi} f^* - \mathcal{T}^{\Pi} f^*\|_d = \|\sum_{s'} (P(s'|s,a) - \hat{P}(s'|s,a)) \max_{\pi \in \Pi} f^*(s', \pi)\|_d \tag{66}$$

$$= \|\sum_{s'} (P(s'|s,a) - \hat{P}(s'|s,a)) f^*(s', \pi^*)\|_d \tag{67}$$

$$\leq C^* \sup_{\pi \in \Pi} \sum_{s_0, a_0} d(s_0, a_0) \mathbb{E}_{a \sim \pi(a|s_0)} \|\hat{\mathcal{T}}^{\Pi} f^* - \mathcal{T} f^*\|(s_0, a) \tag{68}$$

$$\leq C^* \sup_{\pi \in \Pi} \sum_{s_0, a_0} d(s_0, a_0) \mathbb{E}_{a \sim \pi(a|s_0)} \|\hat{\mathcal{T}}^{\Pi} \hat{f}_k - \mathcal{T} \hat{f}_k\|(s_0, a) \tag{69}$$

The last inequality holds because the assumption that the optimal policy is reliable, so its value function would have an ideally low uncertainty.

Therefore, we have,

$$\|\hat{f}_k - f^*\|_d \leq \sup_{\pi \in \Pi} \sum_{s_0} d(s_0) \zeta(s_0, \pi) + \tag{70}$$

$$\mathcal{O}\left( \sup_{\substack{\pi \in \Pi, T \geq 0 \\ s_0, a_0, d(s_0, a_0) > 0}} \mathbb{E}_{s_T, a_T \sim P(s_T, a_T | \pi, s_0, a_0)} (\sum_{t=0}^{T-1} \gamma^{t+1} \zeta_{t+1} - \gamma^t \zeta_t)^2 \right) \tag{71}$$

Completing the proof.

Theorem 2 inspires us that when restricting the policy candidate set, it is essential not only to constrain the uncertainty of the new policy's actions, as in traditional pessimistic algorithms, but also to limit the growth tendency of the Bellman uncertainty caused by the new policy. By controlling this tendency to be as minimal as possible, even ensuring that the Bellman uncertainty monotonically decreases over time steps, we can guarantee that the learned value function exhibits better performance and consequently induces a policy with superior performance.

**Proposition 3.** *If the first term of Eq.(12) is bounded, i.e., $\forall \pi \in \Pi$, we have $\mathbb{E}_{d(s_0)} \zeta_{\hat{f}_k}(s_0, \pi) \leq c$, then we can bound the second term with one-step Lyapunov Uncertainty-penalization, i.e., $\forall s \sim \mathcal{D}$,*

$$\min_{\pi} [\gamma \mathbb{E}_{P(s'|s,a)} \zeta_{\hat{f}_k, t+1}(s', \pi) - \zeta_{\hat{f}_k, t+1}(s, \pi)] \tag{72}$$

$$\Rightarrow \min_{\pi} \mathbb{E}_{P(\tau_T | \pi, s_0 = s)} (\sum_{t=0}^{T-1} [\gamma^{t+1} \zeta_{\hat{f}_k, t+1} - \gamma^t \zeta_{\hat{f}_k, t}] \tag{73}$$

*Furthermore, if we assume the dataset fully covers dynamics modes, i.e., $\forall s, a \in \mathcal{D}, P(s'|s,a) \subseteq \hat{P}(s'|s,a)$, then the left part could be controlled by Lyapunov value estimation.*

*Proof of Proposition 3.* $\min_{\pi} \mathbb{E}_{a \sim \pi(a|s)} \zeta_f(s, a)$; This constrains the new policy would not generated OOD actions beyond the demonstration of the offline data, i.e., $supp(\pi(a|s)) \subset supp(\pi_\beta(a|s))$. In previous works Wu et al. (2022); Mao et al. (2024), such supported constraint is often achieved in a data density based way as $\min_{\pi} \sum_{a \notin \pi_\beta} \pi(a|s)$. Then we will show that the current step's error controlling minimizes the upper bound of the above supported constraint,

$$\min_{\pi} \sum_{a \notin \pi_\beta} \pi(a|s) \Leftrightarrow \max_{\pi} \mathbb{E}_{a \in \pi(a|s)} d(s, a) \tag{74}$$

$$\leq^{(a)} \max_{\pi} C \cdot \mathbb{E}_{a \in \pi(a|s)} \frac{1}{\zeta_f^2(s, a)} \tag{75}$$

$$\Leftrightarrow \min_{\pi} \mathbb{E}_{a \in \pi(a|s)} \zeta_f^2(s, a) \tag{76}$$

$$\Leftrightarrow^{(b)} \min_{\pi} \mathbb{E}_{a \in \pi(a|s)} \zeta_f(s, a) \tag{77}$$

The inequality $(a)$ holds because of the Lemma 1 in Appendix A.2. The equivalence $(b)$ holds because the Bellman uncertainty is always non-negative.

Then with this constraint, we can have the new policy would reject the actions that is not supported by the behavior policy. This means, the new policy would only select the data-supported actions. Then,

$$\min_\pi \mathbb{E}_{P(\tau_T|\pi,s_0=s,a_0=a)}\left(\sum_{t=0}^{T-1}[\gamma^{t+1}\zeta_{\hat{f}_k,t+1} - \gamma^t\zeta_{\hat{f}_k,t}]\right) \tag{78}$$

$$\leq^{(a)} \min_\pi \frac{1-\gamma^T}{1-\gamma} \max_{s_t\in P(s_t|\pi,s_0=s)} (\gamma\mathbb{E}_{P(s_{t+1}|s_t,\pi)}\zeta_{\hat{f}_k}(s_{t+1},\pi) - \zeta_{\hat{f}_k}(s_t,\pi)) \tag{79}$$

$$\leq^{(b)} \min_\pi \frac{1-\gamma^T}{1-\gamma} \max_{s_t\in P(s_t|\pi_\beta,s_0=s)} (\gamma\mathbb{E}_{P(s_{t+1}|s_t,\pi)}\zeta_{\hat{f}_k}(s_{t+1},\pi) - \zeta_{\hat{f}_k}(s_t,\pi)) \tag{80}$$

$$\leq^{(c)} \min_\pi \frac{1-\gamma^T}{1-\gamma} \max_{s\in\mathcal{D}}(\gamma\mathbb{E}_{P(s'|s,\pi)}\zeta_{\hat{f}_k}(s',\pi) - \zeta_{\hat{f}_k}(s,\pi)) \tag{81}$$

The inequality $(a)$ is obtained using the formula for the sum of a geometric series. The inequality $(b)$ is due to $\pi \subseteq \pi_\beta$, where $\pi_\beta$ is the behavior policy. Finally, inequality $(c)$ holds because $P(s_t|\pi_\beta, s_0 = s) \subseteq \mathcal{D}$. Then we have, $\forall s \in \mathcal{D}$,

$$\min_\pi(\gamma\mathbb{E}_{P(s'|s,\pi)}\zeta_{\hat{f}_k}(s',\pi) - \zeta_{\hat{f}_k}(s,\pi)) \Rightarrow \min_\pi \mathbb{E}_{P(\tau_T|\pi,s_0=s)}\left(\sum_{t=0}^{T-1}[\gamma^{t+1}\zeta_{\hat{f}_k,t+1} - \gamma^t\zeta_{\hat{f}_k,t}]\right) \tag{82}$$

Then we would bound the left part through the Lyapunov value estimation. Due to $\pi(a|s) \propto \hat{f}_k(s,a)$, then the penalization to the $s,a$ pairs is equivalent to minimize the preference of $\pi$ to the action $a$ at state $s$. In this way, the Lyapunov Uncertainty-penalization could be converted to,

$$\min_\pi(\gamma \max_{s'\in\hat{P}(s'|s,\pi)} \zeta_{\hat{f}_k}(s',\pi) - \zeta_{\hat{f}_k}(s,\pi)) \geq \min_\pi(\gamma \max_{s'\in P(s'|s,\pi)} \zeta_{\hat{f}_k}(s',\pi) - \zeta_{\hat{f}_k}(s,\pi)) \tag{83}$$

$$\geq \min_\pi(\gamma\mathbb{E}_{P(s'|s,\pi)}\zeta_{\hat{f}_k}(s',\pi) - \zeta_{\hat{f}_k}(s,\pi)) \tag{84}$$

The first inequality holds because of $P(s'|s,a) \subseteq \hat{P}(s'|s,a)$. Completing the proof.

# B ADDITIONAL EXPERIMENTAL DETAILS

## B.1 TESTING ON ENVIRONMENTS WITH OOD OBSERVATIONS

In this section, we conducted tests on benchmarks with OOD observations. This type of testing is primarily aimed at evaluating the generalization ability of conservative/pessimistic methods in offline reinforcement learning when there may be some noise present in the environment to make the agent deviate from the in-distributional regions. To address this challenge, we compared the performance of the proposed LUC method with RORL (Yang et al., 2022) and OSR (Jiang et al., 2023) methods. These two methods are also improvements upon traditional conservative methods, incorporating enhancements such as smoothness constraints (RORL) and recovery constraints (OSR) to enhance the generalization performance of models on unknown states. We utilize the models trained on the 'medium' datasets of the three benchmarks - 'Halfcheetah', 'Hopper' and 'Walker2d', and three kinds of OOD noises - 'random', 'action_diff' and 'min_Q' like in (Yang et al., 2022).

The results are shown in the Figure 5. We can observe that the proposed LUC method achieved good results on most benchmarks, particularly on the two types of adversarial attacks related to action differences. This may be because the learning approach in this paper has the weakest reliance on the behavioral policy, as the conservatism of LUC mainly stems from the evaluation of consequential reliability rather than a specific action distribution. In the context of min_Q, this tests the robustness of these methods in maintaining the optimality of the value function. In this type of attack, LUC also exhibited superior performance compared to other methods, indicating that LUC has better robustness in dealing with the OOD situations than other methods.

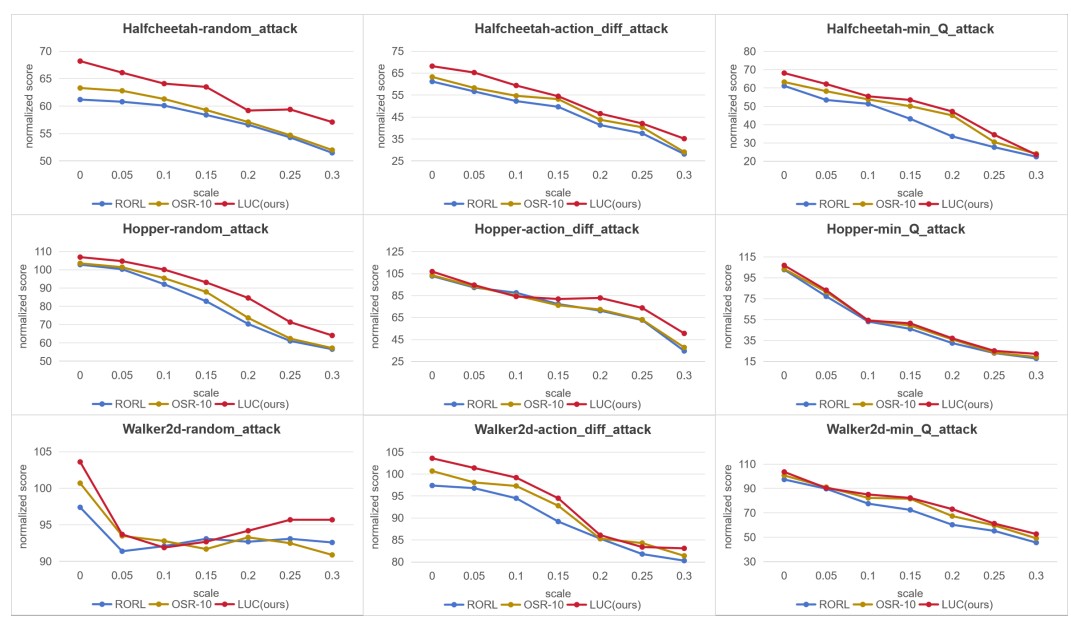

Figure 5: Results of RORL, OSR and LUC on environments with OOD observations.

## B.2 CODE

We constructed our approach based on the RORL project available on GitHub[3]. The rationale behind selecting YangRui2015's project is as follows: 1) The RORL framework serves as a fundamental benchmark for conservative offline reinforcement learning, built on the PBRL implementation (Bai et al., 2022). 2) Implementing conservative Q functions is straightforward with the RORL framework. 3) As far as we are aware, the RORL framework stands out as the leading baseline in MuJoCo benchmarks. The code for our approach is included in the supplementary material.

## B.3 CONSTRUCTION OF OUT-OF-DISTRIBUTION MUJOCO BENCHMARKS

In this section, we introduce how to construct the testing environments for Out-of-distribution MuJoCo benchmarks in detail. First, we set three kinds of perturbations (different scales and intervals) over three kinds of MuJoCo environments as shown in Table 5. The perturbation is randomly sampled from the Uniform distribution.

Table 5: Parameters for the construction of Out-of-distribution MuJoCo benchmarks.

|  | Halfcheetah | | | Hopper | | | Walker2d | | |
|---|---|---|---|---|---|---|---|---|---|
|  | small | medium | large | small | medium | large | small | medium | large |
| scales | 0.05 | 0.15 | 0.3 | 0.01 | 0.03 | 0.05 | 0.03 | 0.05 | 0.07 |
| intervals | 10 | 50 | 100 | 100 | 100 | 100 | 10 | 50 | 100 |

Then we visualize some of the perturbed situations, as is shown in Figure 6.

## B.4 HYPERPARAMETERS OF LUC

In Table 6 and Table 7, we give the hyperparameters used by LUC to generate Table 1 results. The $\lambda_{LUC}$ is the weight of the reward shaping.

---

[3]Project of RORL: https://github.com/YangRui2015/RORL

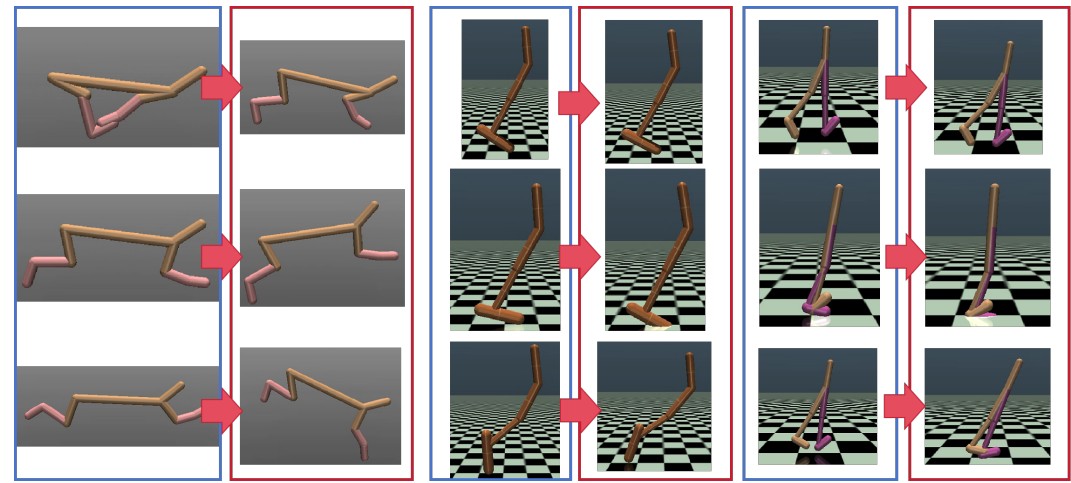

Figure 6: Some visualized samples of perturbations. The samples in blue box is the normal states, while the red box is the corresponding perturbed states. First line is the small scales of perturbation; second line is medium; third line is large.

Table 6: Hyperparameters of LUC in standard MuJoCo benchmarks.

|  | Halfcheetah | Hopper | Walker2d |
|---|---|---|---|
| $\lambda_{LUC}$ | 0.1 | 0.1 | 0.05 |

Table 7: Hyperparameters of LUC in adversarial attack benchmarks.

|  | Halfcheetah | Hopper | Walker2d |
|---|---|---|---|
| $\lambda_{LUC}$ | 0.1 | 0.1 | 0.1 |

## B.5 NEURAL NETWORK STRUCTURES OF LUC

In this section, we introduce the structure of the networks we use in this paper: policy network and Q network.

The structure of the policy network and Q networks is as shown in Table 8, where 's_dim' is the dimension of states and 'a_dim' is the dimension of actions. 'h_dim' is the dimension of the hidden layers, which is usually 256 in our experiments. The policy network is a Guassian policy and the Q networks includes ten Q function networks and ten target Q function networks.

Table 8: The structure of the policy net and the Q networks.

| policy net | Q net |
|---|---|
| Linear(s_dim, 256) | Linear(s_dim, h_dim) |
| Relu() | Relu() |
| Linear(h_dim, h_dim) | Linear(h_dim, h_dim) |
| Relu() | Relu() |
| Linear(h_dim, a_dim) | Linear(h_dim, 1) |

## B.6 COMPUTE RESOURCES

We conducted all our experiments using a server equipped with one Intel Xeon Gold 5218 CPU, with 32 cores and 64 threads, and 256GB of DDR4 memory. We used a NVIDIA RTX3090 GPU

with 24GB of memory for our deep learning experiments. All computations were performed using Python 3.8 and the PyTorch deep learning framework.

## C  DISCUSSION

### C.1  LIMITATIONS

In highly stochastic MDP environments with incomplete dataset coverage of transition outcomes, this paper's method may increase the likelihood of the agent straying from low-uncertainty regions, compromising decision reliability. Nevertheless, experimental results in Section 5.3 highlight the superior reliability of our LUC method over alternative approaches, showcasing enhanced generalization abilities in addressing previously unseen OOD scenarios.

### C.2  DIFFERENCES BETWEEN LUC AND ROBUST RL

Robust RL methods, although they seem to share a similar form to our method (Lyapunov uncertainty control), it is crucial to note the fundamental differences between them and some technical tips:

1) Objective and motivation. Robust RL still falls short of addressing the safety issue we mentioned before. Specifically, the pessimism of Robust RL stems from penalizing with uncertainty in the outcome predictions of actions, to deal with the problem of distributional shift. However, if some behaviors that could lead to the deviation from the safe regions are supported well by dataset, the penalty loses its effectiveness due to that the uncertainty in outcome predictions would be tiny, thereby exacerbating the risk of entering high-uncertainty regions. While in our method, we aim to learn the policy to satisfy the Lyapunov reliable properties, of which we has shown the effectiveness in stable safe control in both theoretical and experimental ways.

2) Penalization mechanism. Robust RL's penalty is reward-driven, aiming to maximize the expected cumulative return under the worst case to deal with transitioned distributional shift. On the other hand, our method places greater emphasis on the safety of the agent's decisions, aiming to minimize the risk of deviation from the safe regions under the worst case to achieve stable safety control for the agent. From this view, we can conclude that the Robust methods focus on the agent's generalization performance in solving offline RL problems, while our method prioritizes stable control of the agent to meet safety requirements in practical applications.

### C.3  DIFFERENCES BETWEEN LUC AND EDAC

There are several key differences between our method, LUC, and the EDAC (An et al., 2021) method in terms of motivation, implementation, and effectiveness, summarized as follows:

1) Motivation: The motivation behind the EDAC method lies in enhancing the sensitivity of Q-ensembles to out-of-distribution (OOD) data. Therefore, the EDAC could be seen as a traditional pessimistic method. Pessimistic methods that focus only on having the agent behave like the demonstrations present in the dataset - while helpful in the quantification for the OOD data, do not fully meet the safety requirements as mentioned before. This is because these methods often focus on an average effect, biasing the agent towards regions with higher data coverage or better model performance, without strictly constraining the agent's activities within the safety region. This can be observed from Theorem 1 in this paper, where both rules defined in Definition 3 are essential for Lyapunov reliability: controlling the action uncertainty of current step and the growth of uncertainty in the future. Unfortunately, previous pessimistic methods only control the former, while ignoring the problem of uncertainty accumulation, failing to satisfy the Lyapunov reliability. While in our method, Q-ensembles serve as a tool to help us measure the safety of the model at certain data points, scoping a safe region for the agent to operate within. These two methods address completely different issues.

2) Implementation: While the EDAC method, similar to traditional pessimistic methods, primarily penalizes the agent for selecting OOD actions, our method not only penalizes such actions but also considers whether choosing a specific action would lead to an increase in uncertainty, thereby preventing the agent from entering regions of high uncertainty (unreliability).

3) Effectiveness: Experimental results indicate that compared to traditional pessimistic methods like EDAC (such as PBRL, RORL, etc.), our method demonstrates greater reliability when facing highly stochastic real-world environments (see Section 5 of our paper).

### C.4 DIFFERENCES BETWEEN LUC AND CQL

When comparing our method to conservative approaches like CQL (Kumar et al., 2020), several key differences can be identified:

1) OOD quantification: Conservative methods like CQL lack the ability to quantitatively measure the degree of out-of-distribution (OOD) data, leading to the rejection of all unseen data and reducing their generalization capabilities. In contrast, our method evaluates OOD data based on uncertainty, assessing the reliability of the model at that specific OOD data point, thus enabling it to generalize effectively on OOD data.

2) Safety assessment of consequences/long-term trends: CQL still focuses on penalizing OOD actions without considering the outcomes' safety of these actions, making it challenging to ensure the agent's operation regions in practical deployment and potentially resulting in trajectory deviations (Zhang et al., 2022; Jiang et al., 2023; Kang et al., 2022). On the other hand, our method defines a closed region where the agent can operate stably, suitable for scenarios requiring higher reliability and safety in decision-making, such as autonomous driving and healthcare applications. This could also be seen in the experimental results in our paper, where we have compared our method LUC with CQL from various benchmarks.

