# OpenReview forum: "Towards Reliable Offline Reinforcement Learning via Lyapunov Uncertainty Control"
_ICLR.cc/2025/Conference — ICLR 2025 Conference Withdrawn Submission_

### Official Review · Reviewer_p3As · 2024-11-02

**Soundness:** 3
**Presentation:** 2
**Contribution:** 3
**Rating:** 6
**Confidence:** 3

**Summary:**

This paper proposes a new approach for offline RL based on Lyapunov stability and Bellman error as a metric of uncertainty, and presents both theoretical and empirical results demonstrating the efficacy of the proposed approach.

**Strengths:**

Originality
- Paper uses Lyapunov stability to improve the performance of offline RL algorithms, which is a relatively novel combinations of existing tools
Quality
- Experiments are thorough, including evaluations on full D4RL suite of tasks, as well as additional experiments investigating OOD states and ablations to the algorithm
- Theoretical results show that proposed approach is conceptually sound, and highlights how errors in different components of the algorithm influence downstream performance
Clarity
- The paper could use some clarifications as discussed below.
Significance
- Experiments show that the proposed approach outperforms some existing approaches to offline RL

**Weaknesses:**

The paper could be further improved if it included comparisons to other standard offline RL, and if it provided a bit more discussion in text explaining each theorem and its significance.

Another paper that is related: In-Distribution Barrier Functions: Self-Supervised Policy Filters that Avoid Out-of-Distribution States

**Questions:**

No questions.

---

> ### Author Response · Authors · 2024-11-19
> **Rebuttal to Reviewer p3As (1/2)**
>
> Thank you for your recognition and the effort you have put into our work. Based on your suggestions, we provide the following clarifications and additions.
>
> **W1... comparisons to other standard offline RL... a bit more discussion in text explaining each theorem and its significance.**
>
> Per your suggestion, we give additional discussions and clarifications about our method from both theoretical and empirical ways.
>
> ** Theoretical clarifications and discussions **
>
> ** Results in 4.1 **
> The main theoretical results in Section 4.1 are used for problem formulation and functional requirements of the method - Definition 1 defines a safe region for the agent to operate,  and Definition 2 defines reliable policy based on it, which is able to verify the safety requirements, i.e., operate within the region defined by Definition 1; Definition 3 and Theorem 1 indicate what kind of policies need to be learned to ensure that the agent can operate stably within the safe region without exceeding it. Proposition 1 demonstrates the existence of policies that meet these functional requirements, validating the feasibility of the method proposed in this paper.
>
> ** Results in 4.3 **
>
> The theoretical results in Section 4.3 (Theorem 2) primarily show the following points:
>
> 1) The method's role in enhancing the agent's performance - it helps optimize the lower bound of the agent's performance. Specifically, our theory attempts to illustrate that when iterated using the empirical Bellman operator, the difference between the learned value function and the true optimal value function, which is also known as the performance lower bound of the offline algorithm, can be controlled through Lyapunov Uncertainty Control. The lower the Lyapunov Uncertainty Control loss, the closer the algorithm's output Q value function is to the optimal value function.
>
> 2) One intuitive way to understand our theoretical result (Theorem 2) is that we control the right term in Theorem 2,
>
> $\frac{C^*}{1-\gamma}\cdot\sup_{\pi\in\Pi}\sum_{s_0}d(s_0)\zeta_{\hat{f}_k}(s_0,\pi) $
>
> $+O(\sup_{{\pi\in\Pi,T\geq 0,s_0,a_0,d(s_0,a_0)> 0}} $
>
> $E_{P(\tau_T|\pi, s_0,a_0)}(\sum_{t=0}^{T-1}[\gamma^{t+1}\zeta_{\hat{f}_k,t+1}$
>
> $ - \gamma^t\zeta_{\hat{f}_k, t}])^2)$.
>
> ,by adjusting the policy candidate set, thereby enhancing the lower bound of the method's output policy performance. Specifically, we align the policy candidate set $\Pi_{Lya}$ with the definition of Lyapunov reliable policy (Definition 3) through the loss function in Eq. (5) (as proven by Proposition 2),
>
> $L_{LUC}(s,a,s',a', f) = E_{a\sim\pi(\hat{a}|s)}\zeta_f(s,\hat{a}) + \beta\cdot \left(\zeta_f(s',a') - \zeta_f(s,a)\right)$.
>
> Such operation can control the right term in Theorem 2.
> And then consequently improve the lower bound of the algorithm's performance.
>
> 3) The proposed method would not conflict with the objective of approaching the optimal policy, under the assumption of optimal coverage (Assumption 1 in the paper).
>
> Finally, we briefly discuss about the theoretical comparison between our method and a classic pessimistic offline RL framework [1].
>
>
> **...Please see next window...**

---

> ### Author Response · Authors · 2024-11-19
> **Rebuttal to Reviewer p3As (2/2)**
>
> Firstly, we construct the Lyapunov policy candidate set according to the Definition 3 in our paper, as,
>
> $\Pi_{Lya}=\{\pi| \forall s\in D, \zeta_f(s,\pi)\leq C_1, \max_{s'\in P(s'|s,\pi)}\zeta_f(s',\pi)\leq \zeta_f(s,\pi) + C_2\}$
>
> where $C_1,C_2$ are two loose coefficient. Then we aim to prove the optimal policy $\pi^*$ in the constructed Lyapunov policy candidate, under certain modification of $C1$ and $C_2$. If we set $C_1=\gamma R_{max}\sqrt{\frac{2}{C^*}\log(\frac{|S||A|2
> ^{|S|}}{\delta})}$, then by the assumption that $sup_{s,a}\frac{d^*(s,a)}{d(s,a)}\leq C^*$, we have,
>
> $\forall s\in D, \zeta_f(s,\pi^*)\leq C_1$. (by plug in the $C$ into Lemma 3 in our paper)
>
> Then again by the assumption that $sup_{s,a}\frac{d^*(s,a)}{d(s,a)}\leq C^*$, we have,
>
> $\forall s, d^*(s)\leq C^*d(s)$
>
> $\Rightarrow \forall s, d^*(s)>0 \rightarrow d(s)>0 \Rightarrow supp(P(s'|s,\pi^*))\subset supp(d)$.
>
> Besides, we have $\pi^*(s)\subset supp(\pi_\beta(a|s))$, where $\pi_\beta$ is the empirical behavior policy according to the dataset, $supp(X)$ is the support set of $X$.
>
> This means that $\max_{s'\in P(s'|s,\pi^*)}\zeta_f(s',\pi^*)\leq C_1$. Denote the maximum uncertainty over the dataset as $\zeta_{max}$ (this is easy to calculate in practice), if we set $C_2 = C_1 - \zeta_{max}$ we have $\forall s,a\in D, C_2 + \zeta_f(s,a)\geq C_1$. Then we have $C_2 + \zeta_f(s,\pi^*)\geq \max_{s'\in P(s'|s,\pi^*)}\zeta_f(s',\pi^*)$. Then the true optimal policy would fall in the Lyapunov reliable policy candidate set $\Pi_{Lya}$ we construct.
>
> Then we can safely plug our method into Corollary 2 from [1]. This corollary is particularly general, as it does not impose any prior assumptions about the policy candidate set, only requiring that the optimal policy is included within that set. Denote the policy learned by LUC as $\hat{\pi}_{Lya}$, then we have,
>
> $J(\pi^*) - J(\hat{\pi}_{Lya})\leq$
>
> $ O(\frac{V_{max}\sqrt{C_2}}{1-\gamma}\sqrt{\frac{\log\frac{|F||\Pi_{Lya}|}{\delta}}{N}}$
>
> $ +\frac{\sqrt{C_2(\epsilon_{F,F} + \epsilon_F)}}{1-\gamma})$
>
> where $V_{max}$ is the upper bound of value function, $|F|$ and $|\Pi_{Lya}|$ are the sizes of the approximated q function space and the policy candidate set, $\epsilon_{F,F}$ and $\epsilon_F$ bound the quality of the learned q function, as are assumed in [1].
>
> Compared to the theoretical results in reference [1], the bound of our method, LUC, is tighter. This is because the Lyapunov reliable policy candidate set $|\Pi_{Lya}|$ we constructed is smaller than the policy candidate set $\Pi$ in the reference [1].
>
> [1] Xie T., et al. Bellman-consistent Pessimism for Offline Reinforcement Learning.
>
> ** Experimental additions and discussions **
>
> We have also added the experiments on the AntMaze and part of the Adroit benchmarks, as follows,
>
> (AntMaze)
> | |CQL|SDC|SPOT|LUC(ours)|
> |-|-|-|-|-|
> |umaze|82.6|89.0|93.5|**94.3**|
> |umaze-diverse|10.2|57.3|40.7|**62.1**|
> |medium-play|59.0|71.9|74.7|**80.1**|
> |medium-diverse|46.6|78.7|**79.1**|78.7|
> |large-play|16.4|37.2|35.3|**44.3**|
> |large-diverse|3.2|33.2|36.3|**41.7**|
> |average|36.3|61.2|59.9|**66.9**|
>
> (Adroit)
> | |pen-hu.|pen-cl.|pen-ex.|hammer-hu.|hammer-cl.|hammer-ex.|door-hu.|door-cl.|door-ex.|avg.|
> |-|-|-|-|-|-|-|-|-|-|-|
> |CQL|37.5|39.2|107.0|4.4|2.1|86.7|9.9|0.4|101.5|43.2|
> |PBRL|35.4|**74.9**|137.7|0.4|0.8|127.5|0.1|4.6|95.7|53.0|
> |LUC(ours)|**41.6**|68.2|**142.5**|**7.9**|**4.7**|**130.4**|**7.8**|**8.4**|**103.4**|**57.2**|
>
> From these benchmarks, we observe that our method outperforms the other methods in most benchmarks. In particular, our method performs much better in the "hammer," "door," and "large" maze benchmarks than other methods. This could be attributed to our method's ability to scope a safe region for the agent to stably operate within, thus addressing more complex environmental dynamics.
>
> **W2.Another paper that is related...**
>
> Thank you for your suggestion. We will add the discussions about the relation between this paper and our work in the revised manuscript.

---

> ### Author Response · Authors · 2024-11-23
> **Friendly reminder.**
>
> Dear Reviewer p3As,
>
> We deeply appreciate the effort you have put into our work. Please note that the discussion period is ending soon. If you have any other questions, concerns, or suggestions, please do not hesitate to post them. We are willing to respond to them.
>
> Best regards
>
> Authors

---

> ### Author Response · Authors · 2024-11-29
>
> Dear Reviewer p3As,
>
> As the discussion period extension is also coming to an end, I wanted to send a friendly reminder that I have not yet received feedback from you. If you have other questions and concerns that could further improve the evaluation of our work, please post them, and we are willing to answer them.
>
> Best regards, Authors

---

### Official Review · Reviewer_gPEg · 2024-11-02

**Soundness:** 2
**Presentation:** 3
**Contribution:** 2
**Rating:** 3
**Confidence:** 4

**Summary:**

This paper introduces Lyapunov Uncertainty Control (LUC), a novel offline RL approach that leverages Lyapunov stability and control-invariant sets to keep the agent within a “safe” state space region, reducing Bellman uncertainty and enhancing decision reliability. By controlling uncertainty growth with a Q-ensemble, LUC shows robust performance across MuJoCo tasks, particularly excelling in out-of-distribution scenarios and improving policy reliability in high-uncertainty offline RL environments.

**Strengths:**

​1.Innovative Approach to Stability Control: The application of Lyapunov instability control to find reliable regions introduces a unique and theoretically grounded approach to managing uncertainty in offline reinforcement learning. This paper’s transition from a density-based to a performance-based approach in utilizing Lyapunov instability is a commendable advancement. By focusing on performance rather than traditional density measures, this method offers a novel, theoretically grounded strategy for managing uncertainty in offline reinforcement learning.
​2.Strong Performance in Walker2D: Experimental results show that the proposed method achieves strong performance across various environments, with particularly outstanding results on the Walker2D task within the MuJoCo environment. This indicates the approach’s effectiveness in managing complex dynamical tasks, further showcasing its robustness.
​3.Clear Visualization of Uncertainty Resolution: The method effectively visualizes the process of resolving uncertainty in out-of-distribution (OOD) states, offering a clear depiction of its robustness. This capability emphasizes the model’s potential to manage and reduce errors in unfamiliar states, which is essential for achieving safe and reliable offline RL.

**Weaknesses:**

1.Similarity to EDAC’s Regularization Approach: The proposed method’s use of a Q-ensemble for regularization has notable similarities with the Ensemble-Diversified Actor Critic (EDAC) approach, which also employs a Q-ensemble to stabilize Q-value estimation through variance reduction and normalization. Ensemble methods are indeed a common strategy in reinforcement learning to address challenges such as overestimation and uncertainty. However, given the alignment with EDAC’s approach, the novelty of the proposed method’s regularization technique may not be fully distinctive. Specifically, EDAC was designed to diversify ensemble members, thereby enhancing policy stability and improving exploration via variance-based adjustments. If the current method mirrors this approach without introducing a clearly defined improvement or novel mechanism, it may be difficult to argue that the proposed regularization is an advancement rather than a reiteration.
​2.Limited Evaluation Scope: The experiments presented in this study are limited to the MuJoCo benchmark, a commonly used environment for continuous control tasks in reinforcement learning. However, focusing solely on MuJoCo may restrict the conclusions that can be drawn about the generalizability and robustness of the proposed approach. Given the objective of establishing reliable offline reinforcement learning policies under uncertainty, it would be beneficial to test the method in more complex and varied environments such as AntMaze or Android-based tasks. These environments present unique challenges, including navigation in maze-like structures and the nuanced action requirements of robotics, which are highly relevant to evaluating an RL method’s capacity to operate in uncertain or OOD conditions. Testing across such diverse benchmarks would significantly strengthen the claims of robustness and adaptability, providing evidence that the proposed method is not overfit to the dynamics and characteristics of MuJoCo tasks.
3.Absence of Direct Comparison to Conservative Offline RL Approaches: Although the study emphasizes stability through Lyapunov principles, it lacks a direct and systematic comparison with conservative offline reinforcement learning techniques, notably Conservative Q-Learning (CQL). CQL has been widely studied for its conservative penalty approach to mitigating out-of-distribution (OOD) risks in offline settings by favoring actions within the distribution of the offline data. Given that the Lyapunov-based uncertainty control also aims to limit the policy’s operations to low-uncertainty (and implicitly, familiar) regions, a comparison with CQL would be invaluable in assessing whether the proposed approach offers any distinct advantage or improvement over conservative penalties in managing OOD actions. Specifically, such a comparison could clarify whether the Lyapunov-based framework introduces any measurable improvements in stability, robustness, or data efficiency compared to CQL and other conservative techniques. A comparative study would also allow for a more nuanced discussion on the benefits and trade-offs between a Lyapunov-centered approach and traditional conservative penalties, potentially uncovering limitations or situations where one may outperform the other. In the absence of this comparative analysis, it remains challenging to fully evaluate the added value and practical impact of the proposed methodology within the broader landscape of offline RL.

**Questions:**

​1.Difference from EDAC’s Q-Ensemble: Could you provide further clarification on how the Q-ensemble utilized in this work differs from that in EDAC, especially in terms of its impact on policy stability and uncertainty control?
​2.Generalizability Beyond MuJoCo: Have you considered extending the experiments to additional environments such as AntMaze or Android-based tasks? Results on these benchmarks could shed light on the method’s generalizability across different reinforcement learning challenges.
​3.Relation to CQL in OOD Scenarios: The proposed method appears to have parallels with conservative Q-learning (CQL) in its handling of OOD data. Could you elaborate on the specific distinctions or similarities between this approach and CQL, particularly regarding their mechanisms for addressing distributional shifts?

---

> ### Author Response · Authors · 2024-11-19
> **Rebuttal to Reviewer gPEg (1/2)**
>
> Thank you for your concerns and the effort you have put into our work. Based on your suggestions, we provide the following clarifications and additions.
>
> **W1/Q1: Difference with EDAC.**
>
> Thank you for your valuable feedback. There are several key differences between our method, LUC, and the EDAC method in terms of motivation, implementation, and effectiveness, summarized as follows:
>
> 1) Motivation: The motivation behind the EDAC method lies in enhancing the sensitivity of Q-ensembles to out-of-distribution (OOD) data. Therefore, the EDAC could be seen as a traditional pessimistic method. (more detailed discussion please see the **reply to Reviewer CW6R's W1**) Pessimistic methods that focus only on having the agent behave like the demonstrations present in the dataset - while helpful in the quantification for the OOD data, do not fully meet the safety requirements as mentioned before. This is because these methods often focus on an average effect, biasing the agent towards regions with higher data coverage or better model performance, without strictly constraining the agent's activities within the safety region. This can be observed from Theorem 1 in this paper, where both rules defined in Definition 3 are essential for Lyapunov reliability: controlling the action uncertainty of current step and the growth of uncertainty in the future. Unfortunately, previous pessimistic methods only control the former, while ignoring the problem of uncertainty accumulation, failing to satisfy the Lyapunov reliability. Additionally, our experimental results, as shown in Table 2, and some references [1][2][3] also support this point.
> While in our method, Q-ensembles serve as a tool to help us measure the safety of the model at certain data points, scoping a safe region for the agent to operate within. These two methods address completely different issues.
>
> 2) Implementation: While the EDAC method, similar to traditional pessimistic methods, primarily penalizes the agent for selecting OOD actions, our method not only penalizes such actions but also considers whether choosing a specific action would lead to an increase in uncertainty, thereby preventing the agent from entering regions of high uncertainty (unreliability).
>
> 3) Effectiveness: Experimental results indicate that compared to traditional pessimistic methods like EDAC (such as PBRL, RORL, etc.), our method demonstrates greater reliability when facing highly stochastic real-world environments (see Section 5.3 of our paper).
>
> Therefore, considering the points above, we believe that our method stands out from traditional pessimistic algorithms like EDAC and holds significant research value.
>
> **W2/Q2: More benchmarks.**
>
> Per your suggestion, we have added the experiments on the AntMaze and part of the Adroit benchmarks, as follows,
>
> (AntMaze)
> | |CQL|SDC|SPOT|LUC(ours)|
> |-|-|-|-|-|
> |umaze|82.6|89.0|93.5|**94.3**|
> |umaze-diverse|10.2|57.3|40.7|**62.1**|
> |medium-play|59.0|71.9|74.7|**80.1**|
> |medium-diverse|46.6|78.7|**79.1**|78.7|
> |large-play|16.4|37.2|35.3|**44.3**|
> |large-diverse|3.2|33.2|36.3|**41.7**|
> |average|36.3|61.2|59.9|**66.9**|
>
> (Adroit)
> | |pen-hu.|pen-cl.|pen-ex.|hammer-hu.|hammer-cl.|hammer-ex.|door-hu.|door-cl.|door-ex.|avg.|
> |-|-|-|-|-|-|-|-|-|-|-|
> |CQL|37.5|39.2|107.0|4.4|2.1|86.7|9.9|0.4|101.5|43.2|
> |PBRL|35.4|**74.9**|137.7|0.4|0.8|127.5|0.1|4.6|95.7|53.0|
> |LUC(ours)|**41.6**|68.2|**142.5**|**7.9**|**4.7**|**130.4**|**7.8**|**8.4**|**103.4**|**57.2**|
>
> From these benchmarks, we observe that our method outperforms the other methods in most benchmarks. In particular, our method performs much better in the "hammer," "door," and "large" maze benchmarks than other methods. This could be attributed to our method's ability to scope a safe region for the agent to stably operate within, thus addressing more complex environmental dynamics.

---

> > ### Author Response · Authors · 2024-11-19
> > **Rebuttal to Reviewer gPEg (2/2)**
> >
> > **W3/Q3: Relation to CQL in OOD Scenarios.**
> >
> > When comparing our method to conservative approaches like CQL, several key differences can be identified:
> >
> > 1) OOD quantification: Conservative methods like CQL lack the ability to quantitatively measure the degree of out-of-distribution (OOD) data, leading to the rejection of all unseen data and reducing their generalization capabilities. In contrast, our method evaluates OOD data based on uncertainty, assessing the reliability of the model at that specific OOD data point, thus enabling it to generalize effectively on OOD data.
> >
> > 2) Safety assessment of consequences/long-term trends: CQL still focuses on penalizing OOD actions without considering the outcomes' safety of these actions, making it challenging to ensure the agent's operation regions in practical deployment and potentially resulting in trajectory deviations [1][2][3]. Similar problem is also seen in pessimistic methods, as is discussed in **the response to Reviewer CW6R's W1**. On the other hand, our method defines a closed region where the agent can operate stably, suitable for scenarios requiring higher reliability and safety in decision-making, such as autonomous driving and healthcare applications. This could also be seen in the experimental results in our paper, where we have compared our method LUC with CQL from various benchmarks.
> >
> > [1] Zhang H, et al. State deviation correction for offline reinforcement learning.
> > [2]Jiang K, et al. Recovering from out-of-sample states via inverse dynamics in offline reinforcement learning.
> > [3]Kang K, et al. Lyapunov density models: Constraining distribution shift in learning-based control.

---

> ### Author Response · Authors · 2024-11-23
> **Friendly reminder.**
>
> Dear Reviewer gPEg,
>
> We deeply appreciate the effort you have put into our work. Please note that the discussion period is ending soon. If you have any other questions, concerns, or suggestions, please do not hesitate to post them. We are willing to respond to them.
>
> Best regards
>
> Authors

---

> ### Author Response · Authors · 2024-11-29
> **Friendly reminder.**
>
> Dear Reviewer gPEg,
>
> As the discussion period extension is also coming to an end, I wanted to send a friendly reminder that I have not yet received feedback from you. I've noticed several misunderstandings regarding my work, and I believe that clarifying these points could significantly improve the evaluation of my submission. Besides, if you have other questions, please post them, and we are willing to answer them.
>
> Best regards, Authors

---

### Official Review · Reviewer_CW6R · 2024-11-09

**Soundness:** 2
**Presentation:** 2
**Contribution:** 1
**Rating:** 3
**Confidence:** 4

**Summary:**

This paper aims to bring the concept of Lyapunov control into offline reinforcement learning. In particular, they aim to tackle the problem of partial data coverage which is very well studied, both theoretically and empirically, in the literature of offline reinforcement learning. The problem of partial data coverage is subsumed under *distributional shift* problem which is a general problem that can be encountered in any learning problem with a fixed dataset, e.g., regression problem, classification problem.

The authors propose a Lyapunov-inspired pessimistic value which also draws inspiration from the work [1]. Some theoretical analysis is provided. In addition, the proposed method is tested empirically on classical MuJoCo tasks.

[1]Ying Jin, Zhuoran Yang, and Zhaoran Wang. Is pessimism provably efficient for offline rl? In
Marina Meila and Tong Zhang (eds.), Proceedings of the 38th International Conference on Machine Learning, ICML 2021, 18-24 July 2021, Virtual Event, volume 139 of Proceedings of Machine Learning Research, pp. 5084–5096. PMLR, 2021

**Strengths:**

1. The structure of the paper is clear.

**Weaknesses:**

1. The contribution is very limited. This is mainly due to the fact the problem formulation in this paper is very partial compared to the current landscape of the offline reinforcement learning.
2. The theoretical results presented in the paper are hard to interpret and compare.
3. The empirical results are not adequate. There are many important benchmarks are left out.

**Questions:**

1. Are authors aware of distributionally robust reinforcement learning (DRRL)? There is a large body of works in this track to tackle distributional shift issue in reinforcement learning, both offline and online. **Fundamentals of DRRL**: [1,2] ;**Offline DRRL**: [3,4] ; **Online DRRL**: [5,6]. This is a very much non-extensive list of works but many of these works shows robustness even when there is complete distribution shift, meaning that the test environment (model) is completely different from the training environment. [3,5] both have extensive simulations on MuJoCo tasks. There is another line of work which specifically tackles state adversarial attack. A notable paper will be [7].

My question is, since you include RORL in your benchmarks, why do you ignore a whole line of robust reinforcement learning works.

2. If you are using pessimism to solve *partial data coverage* issue, you also omitted a seminal work [8]. Could you comment on this paper and compare it with your approach?

3. Is your theoretical guarantees comparable to the ones presented in many other offline RL papers?

References:
[1] Iyengar, G. N. (2005). Robust dynamic programming. Mathematics of Operations Research, 30(2):257–280.
[2] Nilim, A. and El Ghaoui, L. (2005). Robust control of Markov decision processes with uncertain transition matrices. Operations Research, 53(5):780–798.
[3] Panaganti, K., Xu, Z., Kalathil, D., and Ghavamzadeh, M. (2022). Robust reinforcement learning using offline data. Advances in Neural Information Processing Systems (NeurIPS).
[4] Shi, L. and Chi, Y. (2022). Distributionally robust model-based offline reinforcement learning with near-optimal sample complexity. arXiv preprint arXiv:2208.05767.
[5] Zhou, R., Liu, T., Cheng, M., Kalathil, D., Kumar, P., and Tian, C. (2023). Natural actor-critic for robust reinforcement learning with function approximation. In Thirty-seventh Conference on Neural Information Processing Systems.
[6] Li, Y. and Lan, G. (2023). First-order policy optimization for robust policy evaluation. arXiv preprint arXiv:2307.15890.
[7] H. Zhang, H. Chen, C. Xiao, B. Li, M. Liu, D. Boning, and C.-J. Hsieh. Robust deep reinforcement learning against adversarial perturbations on state observations. Advances in Neural Information Processing Systems (NeurIPS), 2020.
[8] Shi, L., Li, G., Wei, Y., Chen, Y., and Chi, Y. (2022). Pessimistic Q-learning for offline reinforcement learning: Towards optimal sample complexity. In Proceedings of the 39th International Conference on Machine Learning, volume 162, pages 19967–20025. PMLR.

---

> ### Author Response · Authors · 2024-11-19
> **Rebuttal to Reviewer CW6R (1/5)**
>
> Thank you for your concerns and the effort you have put into our work. Based on your suggestions, we provide the following clarifications and additions. We believe that we have addressed your concerns and questions in detail, so the rebuttal is a total of **five windows**. If possible, please read it patiently.
>
> **W1: ...contribution...the problem formulation...partial...**
>
> It is worth emphasizing that the core problem investigated in this paper is to enhance the safety of offline reinforcement learning agents. Although it has some association with the issue of partial data coverage, this is not the primary focus. More precisely, our aim is to stop the offline-learned agent from entering areas that could cause severe consequences after deployment. Specifically, in certain applications such as autonomous driving and healthcare, the safety requirements for decision making are extremely strict, demanding that every decision made by the agent at each step be safe.
>
> Meanwhile, current methods like the pessimistic and DRRL methods fall short in handling this issue. For instance,
> Pessimistic methods such as MOPO, PBRL and RORL mainly concentrate on making the agent act in accordance with the demonstrations in the dataset. Although these methods are good at quantifying the OOD data, they may not entirely fulfill the previously mentioned safety requirements. On the other hand, Robust RL methods (e.g., [4],[5]) aim to improve the agent's capacity to deal with distributional shift by establishing the uncertainty set of the transition function and optimizing the lower bound of the policy's long term returns under the worst case scenario within this set. Unfortunately, Robust RL also fails to take into account the safety issue described earlier (more details could be found in the response to **Q1**). In other words, they are unable to prevent the learned agent from straying from the safe region.
>
> By contrast, our approach treats the above safe offline RL problem as a stable control task and incorporates Lyapunov - like constraints, which empowers the agent to operate stably  in the future within the specified safe region. To accomplish this objective, we put forward a novel approach which not only restricts the uncertainty of the agent's current actions but also constrains the accumulation of uncertainty throughout the entire generated trajectory.
>
> To further empirically verify the above point about safety, we attach the experiment on a simple kind of benchmarks based on MuJoCo control suites, which amplifies the feedback of bad decisions from the environment (in the test stage: if reward < 0, reward*=1000). Such benchmarks are more sensitive on the agent's dangerous behaviors, so it puts forward a higher requirement for the safety of agents' behaviors. The results are as follows, where all the agents are trained on 'medium-expert' datasets of D4RL.
>
> | |Halfcheetah|Hopper|Walker2d|
> |-|-|-|-|
> |RORL|87.4|110.4|75.2|
> |LUC(ours)|**92.3**|**112.9**|**83.7**|
>
> From the results above, we can observe that our method LUC outperforms the RORL on the three benchmarks, demonstrating that our method's ability of avoiding those high-risk regions during testing, i.e., stable safe control.
>
> [1] Zhang H, et al. State deviation correction for offline reinforcement learning.
>
> [2]Jiang K, et al. Recovering from out-of-sample states via inverse dynamics in offline reinforcement learning.
>
> [3]Kang K, et al. Lyapunov density models: Constraining distribution shift in learning-based control.
>
> [4]Shi, L, et al. Distributionally robust model-based offline reinforcement learning with near-optimal sample complexity.
>
> [5]Panaganti, et al. Robust reinforcement learning using offline data.

---

> > ### Author Response · Authors · 2024-11-19
> > **Rebuttal to Reviewer CW6R (2/5)**
> >
> > **W2: ...theoretical results...hard to interpret and compare./Q3: ...theoretical guarantees comparable...?**
> >
> > ** Results in 4.1 **
> > The main theoretical results in Section 4.1 are used for problem formulation and functional requirements of the method - Definition 1 defines a safe region for the agent to operate,  and Definition 2 defines reliable policy based on it, which is able to verify the safety requirements, i.e., operate within the region defined by Definition 1; Definition 3 and Theorem 1 indicate what kind of policies need to be learned to ensure that the agent can operate stably within the safe region without exceeding it. Proposition 1 demonstrates the existence of policies that meet these functional requirements, validating the feasibility of the method proposed in this paper.
> >
> > ** Results in 4.3 **
> > The theoretical results in Section 4.3 (Theorem 2) primarily show the following points:
> >
> > 1) The method's role in enhancing the agent's performance - it helps optimize the lower bound of the agent's performance. Specifically, our theory attempts to illustrate that when iterated using the empirical Bellman operator, the difference between the learned value function and the true optimal value function, which is also known as the performance lower bound of the offline algorithm, can be controlled through Lyapunov Uncertainty Control. The lower the Lyapunov Uncertainty Control loss, the closer the algorithm's output Q value function is to the optimal value function.
> >
> > 2) One intuitive way to understand our theoretical result (Theorem 2) is that we control the right term in Theorem 2,
> >
> > $\frac{C^*}{1-\gamma}\cdot\sup_{\pi\in\Pi}\sum_{s_0}d(s_0)\zeta_{\hat{f}_k}(s_0,\pi) $
> >
> > $+O(\sup_{{\pi\in\Pi,T\geq 0,s_0,a_0,d(s_0,a_0)> 0}} $
> >
> > $E_{P(\tau_T|\pi, s_0,a_0)}(\sum_{t=0}^{T-1}[\gamma^{t+1}\zeta_{\hat{f}_k,t+1}$
> >
> > $ - \gamma^t\zeta_{\hat{f}_k, t}])^2$.
> >
> > ,by adjusting the policy candidate set, thereby enhancing the lower bound of the method's output policy performance. Specifically, we align the policy candidate set $\Pi_{Lya}$ with the definition of Lyapunov reliable policy (Definition 3) through the loss function in Eq. (5) (as proven by Proposition 2),
> >
> > $L_{LUC}(s,a,s',a', f) = E_{a\sim\pi(\hat{a}|s)}\zeta_f(s,\hat{a}) + \beta\cdot \left(\zeta_f(s',a') - \zeta_f(s,a)\right)$.
> >
> > Such operation can control the right term in Theorem 2.
> > And then consequently improve the lower bound of the algorithm's performance.
> >
> > 3) The proposed method would not conflict with the objective of approaching the optimal policy, under the assumption of optimal coverage (Assumption 1 in the paper).
> >
> > Finally, we briefly discuss about the theoretical comparison between our method and a classic pessimistic offline RL framework [6].
> >
> > **...please see next window...**

---

> ### Author Response · Authors · 2024-11-19
> **Rebuttal to Reviewer CW6R (3/5)**
>
> Firstly, we construct the Lyapunov policy candidate set according to the Definition 3 in our paper, as,
>
> $\Pi_{Lya}=\{\pi| \forall s\in D, \zeta_f(s,\pi)\leq C_1, \max_{s'\in P(s'|s,\pi)}\zeta_f(s',\pi)\leq \zeta_f(s,\pi) + C_2\}$
>
> where $C_1,C_2$ are two loose coefficient. Then we aim to prove the optimal policy $\pi^*$ in the constructed Lyapunov policy candidate, under certain modification of $C1$ and $C_2$. If we set $C_1=\gamma R_{max}\sqrt{\frac{2}{C^*}\log(\frac{|S||A|2
> ^{|S|}}{\delta})}$, then by the assumption that $sup_{s,a}\frac{d^*(s,a)}{d(s,a)}\leq C^*$, we have,
>
> $\forall s\in D, \zeta_f(s,\pi^*)\leq C_1$. (by plug in the $C$ into Lemma 3 in our paper)
>
> Then again by the assumption that $sup_{s,a}\frac{d^*(s,a)}{d(s,a)}\leq C^*$, we have,
>
> $\forall s, d^*(s)\leq C^*d(s)$
>
> $\Rightarrow \forall s, d^*(s)>0 \rightarrow d(s)>0 \Rightarrow supp(P(s'|s,\pi^*))\subset supp(d)$.
>
> Besides, we have $\pi^*(s)\subset supp(\pi_\beta(a|s))$, where $\pi_\beta$ is the empirical behavior policy according to the dataset, $supp(X)$ is the support set of $X$.
>
> This means that $\max_{s'\in P(s'|s,\pi^*)}\zeta_f(s',\pi^*)\leq C_1$. Denote the maximum uncertainty over the dataset as $\zeta_{max}$ (this is easy to calculate in practice), if we set $C_2 = C_1 - \zeta_{max}$ we have $\forall s,a\in D, C_2 + \zeta_f(s,a)\geq C_1$. Then we have $C_2 + \zeta_f(s,\pi^*)\geq \max_{s'\in P(s'|s,\pi^*)}\zeta_f(s',\pi^*)$. Then the true optimal policy would fall in the Lyapunov reliable policy candidate set $\Pi_{Lya}$ we construct.
>
> Then we can safely plug our method into Corollary 2 from [6]. This corollary is particularly general, as it does not impose any prior assumptions about the policy candidate set, only requiring that the optimal policy is included within that set. Denote the policy learned by LUC as $\hat{\pi}_{Lya}$, then we have,
>
> $J(\pi^*) - J(\hat{\pi}_{Lya})\leq $
>
> $O(\frac{V_{max}\sqrt{C_2}}{1-\gamma}\sqrt{\frac{\log\frac{|F||\Pi_{Lya}|}{\delta}}{N}}$
>
> $ +\frac{\sqrt{C_2(\epsilon_{F,F} + \epsilon_F)}}{1-\gamma})$
>
> where $V_{max}$ is the upper bound of value function, $|F|$ and $|\Pi_{Lya}|$ are the sizes of the approximated q function space and the policy candidate set, $\epsilon_{F,F}$ and $\epsilon_F$ bound the quality of the learned q function, as are assumed in [6].
>
> Compared to the theoretical results in reference [6], the bound of our method, LUC, is tighter. This is because the Lyapunov reliable policy candidate set $|\Pi_{Lya}|$ we constructed is smaller than the policy candidate set $\Pi$ in the reference [6].
>
> [6] Xie T., et al. Bellman-consistent Pessimism for Offline Reinforcement Learning.
>
> **W3: The empirical results are not adequate. There are many important benchmarks are left out.**
>
> Per your suggestion, we have added the experiments on the AntMaze and part of the Adroit benchmarks, as follows,
>
> (AntMaze)
> | |CQL|SDC|SPOT|LUC(ours)|
> |-|-|-|-|-|
> |umaze|82.6|89.0|93.5|**94.3**|
> |umaze-diverse|10.2|57.3|40.7|**62.1**|
> |medium-play|59.0|71.9|74.7|**80.1**|
> |medium-diverse|46.6|78.7|**79.1**|78.7|
> |large-play|16.4|37.2|35.3|**44.3**|
> |large-diverse|3.2|33.2|36.3|**41.7**|
> |average|36.3|61.2|59.9|**66.9**|
>
> (Adroit)
> | |pen-hu.|pen-cl.|pen-ex.|hammer-hu.|hammer-cl.|hammer-ex.|door-hu.|door-cl.|door-ex.|avg.|
> |-|-|-|-|-|-|-|-|-|-|-|
> |CQL|37.5|39.2|107.0|4.4|2.1|86.7|9.9|0.4|101.5|43.2|
> |PBRL|35.4|**74.9**|137.7|0.4|0.8|127.5|0.1|4.6|95.7|53.0|
> |LUC(ours)|**41.6**|68.2|**142.5**|**7.9**|**4.7**|**130.4**|**7.8**|**8.4**|**103.4**|**57.2**|
>
> From these benchmarks, we observe that our method outperforms the other methods in most benchmarks. In particular, our method performs much better in the "hammer," "door," and "large" maze benchmarks than other methods. This could be attributed to our method's ability to scope a safe region for the agent to stably operate within, thus addressing more complex environmental dynamics.

---

> > ### Author Response · Authors · 2024-11-19
> > **Rebuttal to Reviewer CW6R (4/5)**
> >
> > **Q1: .. distributionally robust reinforcement learning ..**
> >
> > Thank you for raising your concerns. Firstly, it is important to emphasize again that our focus is on addressing the safety issues that offline-trained agents face in online deployment - specifically, how to control the agent to operate stably within a safe region. Robust RL methods, although they seem to share a similar form to our method (Lyapunov uncertainty control), it is crucial to note the fundamental differences between them and some technical tips:
> >
> > 1) Objective and motivation. Robust RL still falls short of addressing the safety issue we mentioned before. Specifically, the pessimism of Robust RL stems from penalizing with uncertainty in the outcome predictions of actions, to deal with the problem of distributional shift. However, if some behaviors that could lead to the deviation from the safe regions are supported well by dataset (as is in the case described in Figure 1 of our paper), the penalty loses its effectiveness due to that the uncertainty in outcome predictions would be tiny, thereby exacerbating the risk of entering high-uncertainty regions. While in our method, we aim to learn the policy to satisfy the Lyapunov reliable properties, of which we has shown the effectiveness in stable safe control in both theoretical and experimental ways.
> >
> > 2) Penalization mechanism. Robust RL's penalty is reward-driven, aiming to maximize the expected cumulative return under the worst case to deal with transitioned distributional shift (e.g. [Robust reinforcement learning using offline data] and [Distributionally Robust Model-Based Offline Reinforcement Learning with Near-Optimal Sample Complexity]). On the other hand, our method places greater emphasis on the safety of the agent's decisions, aiming to minimize the risk of deviation from the safe regions under the worst case (see Eq.3 in our paper) to achieve stable safety control for the agent. From this view, we can conclude that the Robust methods focus on the agent's generalization performance in solving offline RL problems, while our method prioritizes stable control of the agent to meet safety requirements in practical applications.
> >
> > 3) Clarification for OOD MuJoCo benchmarks we use. The purpose of introducing this setting is to enhance the transition stochastics of benchmarks, simulating highly stochastic environments in actual deployment to test whether the agent can still operate stably and consistently in safe regions under such conditions, rather than testing the agent's robustness against the environmental perturbation.
> >
> > 4) Why we only choose RORL for comparison? As described earlier, it is evident that the Robust RL method differs significantly from our work in terms of settings and objectives. Consequently, we have chosen to compare only with RORL. This decision is based on two main points: 1) RORL combines the characteristics of both pessimistic learning and robust learning, making it a more representative and effective baseline; 2) The RORL implementation is open-source and easy to implement, facilitating a fairer comparison. Other Robust RL methods either focus on settings that differ from our offline context or do not provide simulation code, which is why we have selected RORL as the main comparison method.

---

> > > ### Author Response · Authors · 2024-11-19
> > > **Rebuttal to Reviewer CW6R (5/5)**
> > >
> > > **Q2: .. a seminal work [8]...**
> > > [8] Shi, L., et al. Pessimistic Q-learning for offline reinforcement learning: Towards optimal sample complexity.
> > >
> > > The innovation of the listed paper [8] lies in automatically updating the (hyper) parameters (e.g. learning rate, ) of pessimistic offline RL algorithms with fine-designed rules during value iteration, ensuring the convergence and performance bound of model-free offline RL algorithms with low sample complexity. It guarantees a similar sample complexity to those model-based offline RL methods like [Policy finetuning: Bridging sample-efficient offline and online reinforcement learning.], while avoiding bias introduced by learning dynamical models. However, from our perspective in addressing the challenges posed by our work, this work [8] has the following issues:
> > >
> > > 1) It remains a traditional pessimistic method. Its core idea still constrains the agent's behavior based on demonstrated actions in the dataset, which, as mentioned in our reply to **W1**, represents an average effect and cannot meet the requirements for stable safety control - i.e., it cannot control the agent's activity within the scoped safe regions.
> > >
> > > 2) It may be challenging to apply in non-tabular data settings, especially in cases of continuous state/action spaces that we focus on. The evidences are: On one hand, as noted in the main conclusion - Theorem 3 of the paper, the bound of the algorithm needs to be determined by the size of the state space S, and in real-world applications, the state space is often infinite, i.e., S is unbounded, rendering the performance bound provided in this paper ineffective. On the other hand, it requires obtaining $max_a Q_h(s_h,a)$ accurately (Algorithm 1-line 13, Algorithm 2-line 20 in this paper), which is difficult to achieve in settings with continuous action spaces. Particularly in offline scenarios, $argmax_a Q_h(s_h,a)$ may be an OOD action, which is hard to predict. In our work, we tend to address complex and highly stochastic control tasks in real-world scenarios, and we believe that in such settings, the method proposed in the paper may not guarantee the performance it claims.
> > >
> > > Considering the above points, we believe that while the method proposed in this paper [8] demonstrates good mathematical properties such as convergence and sample complexity under certain settings, it may struggle to meet the requirements for stable safety control in highly random and continuous control tasks during practical deployment. Our work offers a solution tailored to high-safety demand tasks in real-world scenarios, and we have theoretically and experimentally validated its effectiveness.

---

> ### Author Response · Authors · 2024-11-23
> **Friendly reminder.**
>
> Dear Reviewer CW6R,
>
> We deeply appreciate the effort you have put into our work. Please note that the discussion period is ending soon. If you have any other questions, concerns, or suggestions, please do not hesitate to post them. We are willing to respond to them.
>
> Best regards
>
> Authors

---

> ### Author Response · Authors · 2024-11-29
> **Fridendly reminder.**
>
> Dear Reviewer CW6R,
>
> As the discussion period extension is also coming to an end, I wanted to send a friendly reminder that I have not yet received feedback from you. I've noticed several misunderstandings regarding my work, and I believe that clarifying these points could significantly improve the evaluation of my submission. Besides, if you have other questions, please post them, and we are willing to answer them.
>
> Best regards, Authors

---

### Author Response · Authors · 2024-11-19
**Global rebuttal/outline.**

Dear reviewers and Area Chair,

Hello everyone. First, we would like to thank you all for your questions and suggestions regarding our work. We have posted responses addressing the concerns raised by each reviewer, striving to explain and clarify your points as much as possible. If possible, we kindly ask you to read them.

Once again, we appreciate the time and effort you have all dedicated to our work. In the final days of the discussion phase, we will update our PDF based on the outcomes of our discussions with you.

Then we gives an outline of the rebuttals, as follows,

1) **Contribution/Problem formulation of our work.** See the response to Reviewer CW6R's W1.

2) **Explanation of theoretical results and theoretical comparison with other methods.** See the response to Reviewer CW6R's W2/Q3, p3As's W1.

3) **Discussions about the relationship with other methods.** This part have following aspests:

    3.1) Robust RL. See the response to Reviewer CW6R's Q1.

    3.2) Pessimistic methods. See the response to Reviewer CW6R's Q2.

    3.3) EDAC. See the response to Reviewer gPEg's W1/Q1.

    3.4) CQL. See the response to Reviewer gPEg's W3/Q3.

4) **Additional benchmarks.** See the response to Reviewer CW6R's W3, gPEg's W2/Q2, p3As's W1.

5) **Additional related work.** Reviewer p3As's W2.

---

### Author Response · Authors · 2024-11-25
**Revision of Submission and Final Clarifications**

Dear reviewers and Area Chair,

As the discussion period is ending, we have prepared a revised version of the PDF for this submission, where the revised contents are marked as **blue**. Since there has been no additional discussion beyond the original reviews, we have incorporated the contents of our first-round rebuttals into the paper.

If you have any concerns, please feel free to share them, and we are willing to do our best to address them in the remaining days to ensure that there are no misunderstandings.

Best, Authors

---

### Author Response · Authors · 2024-12-02
**Friendly Reminder: Discussion Period Ending.**

Dear reviewers,

We deeply appreciate the efforts and time you have put into our work. We believe our previous rebuttals have better address your concerns.

As the discussion period ends in less than 24 hours, we look forward to hearing from you if you have additional concerns or questions about our work. If possible, we will attach more technical supplementary material to improve your comments on our work.

Best, Authors

---

### Note · Authors · 2025-01-09

I have read and agree with the venue's withdrawal policy on behalf of myself and my co-authors.